# Economic and econometric methods to measure the illicit tobacco trade: A scoping review

Pyi Pyi Phyo[1]*, Natalie Walker[2], Braden Te Ao[1], Erwann Sbai[3], Chris Bullen[1]

1 School of Population Health, Faculty of Medical and Health Sciences, The University of Auckland, Grafton, Auckland, New Zealand, 2 College of Medicine and Public Health, Flinders University, Adelaide, South Australia, Australia, 3 Department of Economics, The University of Auckland. Sir Owen G Glenn Building, Auckland, New Zealand

* hhhk.family@gmail.com

## Abstract

This scoping review aimed to identify suitable economic and econometric methods for measuring the illicit tobacco trade. We searched six key databases for public health and economics papers (PubMed, CINAHL, EMBASE, EconLit, ABI/Inform, and Medline), two economic working paper platforms (SSRN and IDEAS), and grey literature via Google and expert-identified articles. Initial screening was undertaken by all authors, with at least three authors conducting a second screening and final paper selection. We included English-language papers (published from 2010 to July 2023) that applied economic or econometric models to illicit tobacco or related topics. We examined the methods, assessing their strengths and limitations from a health equity perspective, and evaluated their applicability to priority populations (rather than assessing the quality of individual models). The review included 39 studies: 16 applied consumption gap analysis (CGA), and 23 used other economic or econometric models (i.e., Exponentiated, Discrete Choice, Extended Cost–Benefit Analysis and A Static Partial Equilibrium, Consumption, Risk Prediction, A Forward-looking Behavioural, Integrated Micro-Macro Demand, Endogenous Switching Regression, Multiple and Non-Linear Regression, Dynamic Projection, Demand-driven Analytical, Econometric Regressions and Modelling and Two-way Fixed Effects models). CGA was primarily used to estimate the size and trends of the illicit tobacco market, whereas other models assessed and quantified past, existing, or potential behaviours related to engagement with tobacco and other products, including illicit tobacco. Only six of the 39 studies addressed health equity. Measuring the illicit tobacco trade is challenging due to its covert nature, methodological limitations, and scarce high-quality data. Method selection depends on the research objective: CGA is suitable for assessing national market trends but is limited in evaluating subpopulations or future policy impacts. Other non-CGA-based economic and econometric models are better for analysing or predicting user behaviour, including from a health equity perspective. **Implications for public health:** Measuring the illicit tobacco trade is challenging.

**Data availability statement:** All data are in the manuscript and supporting information files.

**Funding:** The authors received no specific funding for this work.

**Competing interests:** I have read the journal's policy, and Pyi Pyi Phyo, Braden Te Ao, and Erwann Sbai have declared that they have no competing interests. Natalie Walker has the following competing interests: Current research funding (salary support) from the NZ Health Research Council and the US National Institute for Health (NIH). Chris Bullen has the following competing interests: Research funding from the NZ Health Research Council, Ministry of Health, Wellcome Trust, and NIH. Financial support from Kenvue SE Asia for travel to an ASEAN smoking cessation network meeting in 2023, 2024 and 2025 and speaking at the WONCA Asia conference in South Korea in 2025.

This review identified a wide variety of economic or econometric methods on this topic and highlighted the need for a greater equity focus when applying these methods. Triangulating findings across the various methods is important moving forward.

## Introduction

The World Health Organization (WHO) defines illicit tobacco trade as "any practice or conduct prohibited by law that relates to the production, shipment, receipt, possession, distribution, sale, or purchase of tobacco, including any practice or act intended to facilitate such activity." [1]. The WHO had estimated that about 10% of all tobacco products consumed globally are illicit [2].

The tobacco industry often highlights the potential for an increase in illicit tobacco as a strategy to delay or reverse the implementation of tobacco control policies, while at the same time engaging in the illicit tobacco trade [3]. For example, in 2024, the newly elected National Party-led coalition New Zealand (NZ) government quickly repealed the previous government's world-leading 'Smokefree Environments and Regulated Products (Smoked Tobacco) Amendment Act', citing, in justification, claims of an increase in the availability of illicit tobacco based on industry-funded studies that estimate that the illicit trade was increasing [4,5]. However, independent analyses using consumption gap analysis (CGA) indicated a fluctuating but downward trend in illicit tobacco in NZ between 2012 and 2023 [6]. According to data collected from NZ Customs, many different types of illicit tobacco were seized during this period – illicit cigarettes, illicit loose tobacco imported from other countries (especially Tonga), chewing tobacco, water pipe tobacco, and domestically grown and produced loose tobacco [7]. Under the NZ Customs and Excise Act 2018, adults are permitted to cultivate up to 5 kg of tobacco annually for personal use, but the sale or distribution of home-grown tobacco is prohibited and illegal [8].

One reason for the different estimates of the size of the illicit tobacco trade is that it is inherently difficult to measure [9]. Researchers and international organisations, such as the World Bank and Tobacconomics, have recommended various methods to measure the illicit tobacco market. These methods range from surveys of people who smoke tobacco (with or without examination of cigarette packs), discarded pack surveys, interviews with key stakeholders (e.g., Customs officials), CGA, and economic modelling (using econometric analysis) [10–12].

CGA examines the difference between estimated tobacco consumption and legal sales [11]. Econometrics uses economic theory, math, and statistics to quantify economic phenomena [13], enabling estimates of how price changes impact illicit tobacco demand. Furthermore, such methods were used to analyse the link between legal sales and factors influencing tobacco demand and smuggling [14].

Economic methods have been widely used in tobacco control research for various purposes, including examining the relationships among taxation, price, consumption, and health outcomes [15,16].These approaches have also been applied to model nicotine addiction within a framework of rational economic behaviour [15,17].In addition, economic models can be used to assess how cigarette taxation may shift demand

for other tobacco and nicotine products through substitution effects by estimating cross-price elasticities and relative price responses [15,16]. Furthermore, economic methods are widely used to evaluate the impact of tobacco control policies on tobacco demand and use [16]. Finally, economic analyses—such as cost–benefit and economic impact analyses—have been employed to assess the overall economic contribution of tobacco [15].In the context of illicit tobacco trade, econometric modelling is used to estimate tax avoidance and evasion by comparing predicted tax-paid sales with observed sales, with the difference interpreted as illicit consumption; this approach has been applied to measure legal cross-border shopping, direct low-tax purchases, and illegal bootlegging in a specific country and to a more limited extent in global and regional estimates [9,15]Generally, it is recommended that multiple methods are used to better understand the scale of, or changes in, the illicit tobacco trade [9].

It remains unclear which economic or econometric methods or models are most appropriate for measuring illicit tobacco trade or use in NZ, an island nation where overall smoking prevalence has declined but substantial inequities persist across population groups [18], particularly in the context of potential implementation of nicotine reduction policy. It is important to examine whether such methods and models consider health equity while measuring illicit tobacco [19].

We undertook a scoping review to identify economic or econometric methods that could be applicable for measuring changes in the extent of the illicit tobacco trade and how these methods could be used to measure differences among various sub-groups of the population, thus promoting a health equity perspective [20]. The focus was particularly relevant in the NZ context and other similar contexts, where the prevalence of tobacco use varies significantly among different ethnic groups and socio-economic strata.

## Methodology

The scoping review protocol was developed in June 2023 and registered with the Open Science Framework (OSF) [21]. The review followed the scoping review guided by the University of South Australia and the Joanna Briggs Institute (JBI) manual for evidence synthesis [22]. We used the criteria outlined in the PRISMA extension for scoping reviews for reporting the review findings [23]. In July 2023, the protocol's literature search strategy was revised, followed by a further revision in May 2024 that integrated feedback from peer reviewers.

### Research questions

The research questions were:

- What economic or econometric methods should be used to measure the size of the illicit tobacco trade?

- What strengths and limitations do these methods have when measuring illicit tobacco use from a health equity perspective?

- Can these methods be used to assess the size and impact of the illicit tobacco market in priority populations (e.g., sub-populations with higher smoking prevalence than the general population, such as Indigenous peoples, and people with low education)?

### Search strategy

The research question followed the population-concept-context framework to determine relevant search terms:

- Population: Active tobacco users

- Concept: Economic or econometric methods

- Context: Global and New Zealand.

The initial search strategy was tested in PubMed, followed by a preliminary screening of studies. The Boolean operator 'AND' was used to link the main search concepts. A 'NOT' operator was then added to refine the results and remove irrelevant articles. An overview of the final search strategy is provided in Table 1.

Five databases (PubMed, CINAHL, EMBASE, EconLit, and ABI/Inform) were searched using the search terms (S1 File, Table A2). Grey literature was searched using Google, following the methods suggested by the University of Otago, New Zealand [24]. The first search was conducted in July-August 2023 and was extended in April 2024 to include Medline and two economic working paper platforms (SSRN and IDEAS) recommended by experts in economics at the University of Auckland (S1 File, Table A3).

## Study selection

Studies were identified, screened, and selected in accordance with PRISMA guidelines [25] (Fig 1) was conducted using the free version of Rayyan Software [26] that allowed multiple researchers to participate online by using the eligibility criteria in Table 2. All five authors screened study titles. Abstracts were then screened by at least three authors, with the final selection made after reviewing the full papers and related appendices. Discussions were held when reviewers disagreed, with a resolution reached by consensus (based on the inclusion criteria (Table 2)).

### Data extraction

Data extraction was undertaken by the first author using Microsoft Excel. The studies were grouped into two categories: those that applied CGA methods and those that used non-CGA methods. Two co-authors reviewed the extracted data. The titles for data extraction and synthesis are presented in Table 3.

### Critical appraisal from a health equity perspective

We examined the strengths and limitations of these methods through a health equity lens, emphasising their relevance to subpopulations, including priority or vulnerable populations, rather than the quality of individual models. Priority populations included, but were not limited to, analyses focusing on specific ethnic groups, gender, socioeconomic status, and mental or physical health characteristics associated with increased vulnerability to tobacco use. Based on Huang et al.'s proof-of-concept quality assessment framework for tobacco models [27], a checklist was used to assess whether studies evaluated the target population, included baseline and policy scenarios, conducted demand analysis, ensured model transparency, and examined equity approaches (S1 File, Table A5). Studies that used CGA were included as a single category in the critical appraisal, as they utilised the same methodological approach.

## Results

### Selection of sources of evidence

Overall, 759 papers were identified, of which 247 were subsequently removed as duplicates. A further 458 articles were excluded after the initial screening because they did not meet the inclusion criteria, leaving 512 records. Another 19

**Table 1. Search strategy.**

| Concept 1 | Concept 2 | Concept 3 | Concept 4 | Concept 5 |
|---|---|---|---|---|
| Illegal | Tobacco products | Econometric/ Economic | Methods | Exclusion |
| illegal OR unlawful OR illicit OR smuggl* OR criminal OR prohibit* OR banned OR contraband | tobacco OR cigarette* OR "roll your own" OR roll-your-own OR "hand rolled cigarette" OR Illicit-Tobacco-Trade | econometric* OR economic* OR tax* OR cost* OR price* OR demand* | model* OR estimat* OR gap-analysis OR experimental-marketplace | NOT drugs OR NOT "substance use" OR NOT children OR NOT adolescent* |

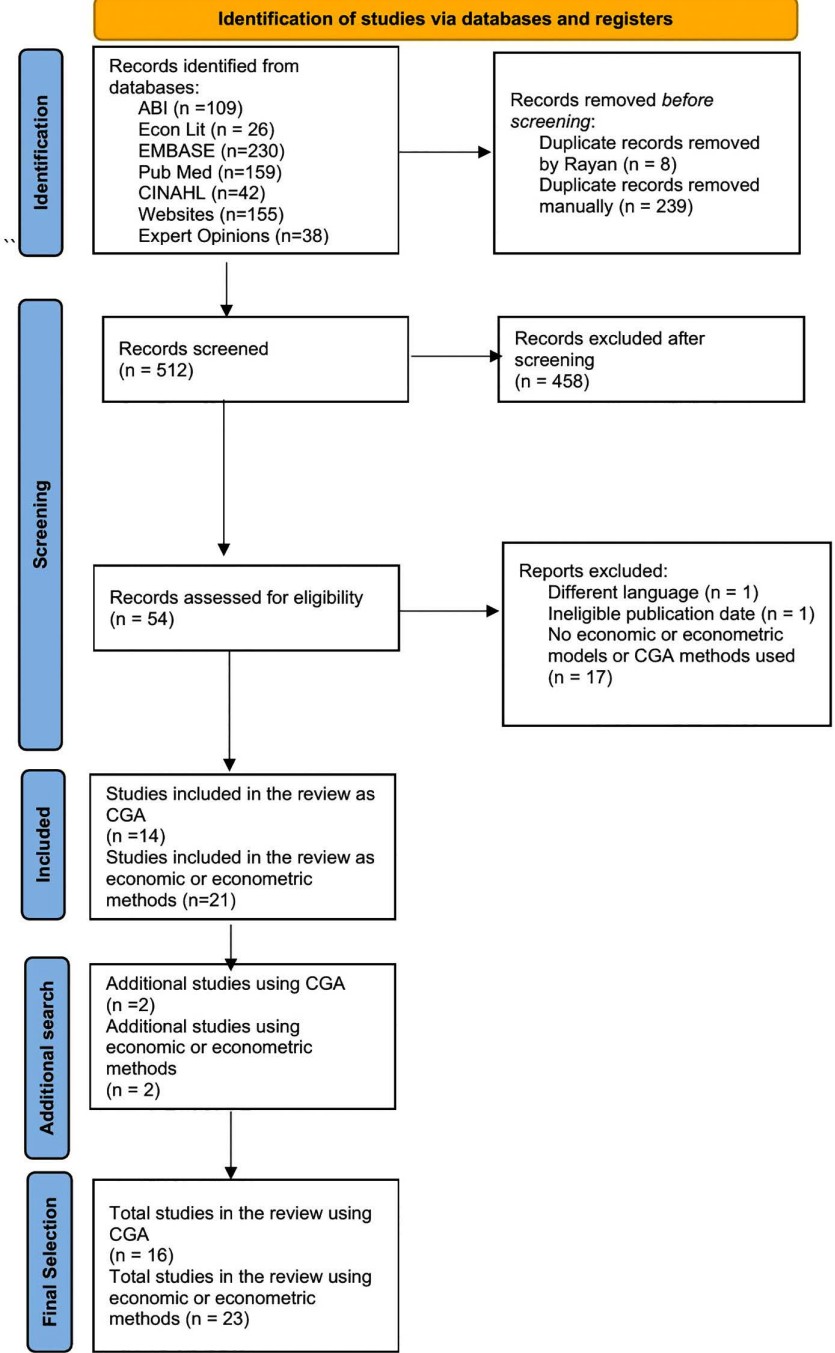

**Fig 1. Flow chart for the study selection using the PRISMA flow diagram.**

studies were removed during the second screening phase: 17 were excluded because they did not use the CGA method or economic or econometric models, one was excluded because it was not published in English, and one was excluded because it was published prior to 2010. Overall, 35 articles underwent full review (Fig 1). Additional searches in MEDLINE, IDEAS, and SSRN yielded four additional papers, bringing the total to 39 studies (Fig 1).

**Table 2. Inclusion and exclusion criteria for the scoping review.**

| Inclusion criteria | Exclusion criteria |
|---|---|
| • All publication types, including systematic reviews, meta-analyses, rapid reviews, scoping reviews, narrative reviews, qualitative research, simulation modelling studies, reports, and guidelines and reports from international organisations (such as the WHO and the World Bank), thesis or dissertations. Since the review examined the methods, study funding by the tobacco industry was not a criterion for exclusion. | • Publications that do not present relevant methodological approaches. |
| • Publications in the English language only. | • Publications in languages other than English. |
| • Publications that examine the economic or econometric models/methods in the illicit tobacco trade. Studies on other forms of illicit trade (e.g., illicit alcohol, illegal drugs, or various types of nicotine and tobacco products) were included, because the methods used in these areas could also be relevant to the illicit tobacco trade, given the similarities between products. | • Publications that do not examine economic or econometric models/methods. |
| • Publications from 2010 to July 2023 (2010 was used as the start date, to capture the most recent methods/models from the past 15 years). | • Exclude articles prior to 1 January 2010, and after 23 July 2023 |

**Table 3. Data extraction and synthesis format.**

| CGA-related studies | Non-CGA-based studies |
|---|---|
| Authors | Authors |
| Year of publication | Year of publication |
| Origin/Country of origin | Origin/Country of origin |
| Aim/Purpose | Aim/Purpose |
| Type of tobacco control policies in the current study context (Existing policy) | Type of tobacco control policies in the current study context (Existing policy) |
| Potential policy scenario (Future policy changes) | Potential policy scenario (Future policy changes) |
| Data analysis and data source | Data collection and data analysis (Models) |
| Unit of measurement | – |
| Comment | Strengths and limitations of models (in a separate table) |

## Characteristics of the studies

Of all studies, 38 were original papers, and one was the editorial. The studies were divided into two groups: CGA-related studies and studies using economic and econometric methods. Sixteen studies used the CGA method. Of these, ten studies were conducted in Asia, three in Africa, two in South America, and one in North America. Twenty-three studies used economic or econometric methods, 12 conducted in North America, five in Europe, three in Oceania, two in South America, and one in Asia.

## Studies using CGA methods

Of the 16 studies using the CGA method (see Table 4 for a detailed synthesis of these studies). All monitored trends in the illicit tobacco trade and estimated its size within a specific country. The CGA method was applied with and without considering tobacco policy changes, allowing the researchers to evaluate trends in consumption patterns under varying

**Table 4. Summary of characteristics of studies that applied CGA methods.**

| Author, Year of publication | Country of origin | Aim/ purpose | Type of tobacco policies in the current study context | Potential policy scenario | Data analysis | Data source | Unit of measurement | Comment | Industry funded |
|---|---|---|---|---|---|---|---|---|---|
| Lavares et al., 2022 | Philippines | To estimate the size of the illicit tobacco market. | Tobacco taxation, including the sin tax, was introduced. | N/A | Cigarette gap or consumption gap, gap or gap) = Legal Sales - Cigarette Consumption | <u>Demand data:</u> Smoking prevalence from the National Nutrition Health Surveys (1998, 2003, 2008, 2013, 2015, and 2018) Population size from the population census <u>Supply Data:</u> Taxed sales data from the Bureau of Internal Revenue. | Number of sticks | Z-test was used for sensitivity analysis. The uplift factor was applied to the calculations generated in 1998. | No |
| Paraje, G., 2019 | Argentina, Brazil, Chile, Colombia, and Peru | To estimate the evolution of the illicit cigarette trade in five South American countries. | N/A | N/A | Total cigarette consumption in a country = registered consumption (tax-paid) + unreported consumption (contraband, tax evasion, tax avoidance, and product counterfeiting) | 1. Argentina: <u>Demand data:</u> National Risk Factors Survey for 2005, 2009, and 2013, and National Survey on the Prevalence of Psychoactive Substance Consumption for 2008 and 2011. <u>Supply Data:</u> Ministry of Agro-industry. 2. Brazil: <u>Demand data:</u> Global Adult Tobacco Survey for 2008 and the National Health Survey 2013. Supply data: Federal Revenue Service. 3. Chile: <u>Demand data:</u> Survey of Drug Use in the General Population for 2008, 2010, 2012, and 2014. 4. Colombia: <u>Demand data:</u> Psychoactive Substance Consumption Survey for 2008 and 2013. 5. Peru: <u>Demand data:</u> National Commission for Development and Life without Drugs for 2006 and 2010. <u>Supply Data:</u> Euromonitor International for Chile, Colombia, and Peru. | Billions of sticks | Produced confidence intervals via bootstrapping. | No |
| Vellios, N. et al., 2019 | South Africa | To assess trends in the size of the illicit cigarette market. | The exercise tax increase was modest, and the tax administration was poor. | N/A | Illicit market size = survey-based consumption - tax-paid consumption  Tax-paid consumption = exercise revenue/exercise tax per cigarette | <u>Demand data:</u> All Media and Products Survey for 2002–2014. National Income Dynamics Study for 2008, 2010, 2012, 2015, and 2017. <u>Supply Data:</u> Annual national budgets by the National Treasury of South Africa. | Billions of sticks | Bootstrapping was conducted with 1000 repetitions to estimate smoking prevalence multiplied by smoking intensity. Utilised an uplift factor to increase population estimate weights. Used underreporting calculations of 5% and 10% for AMPS and 15% and 20% for NIDS. | No |

*(Continued)*

**Table 4.** (Continued)

| Author, Year of publication | Country of origin | Aim/ purpose | Type of tobacco policies in the current study context | Potential policy scenario | Data analysis | Data source | Unit of measurement | Comment | Industry funded |
|---|---|---|---|---|---|---|---|---|---|
| Lam et al., 2018 [+] | Hong Kong | To estimate illicit cigarette tobacco consumption size. | N/A | N/A | Top-down approach: Illicit consumption = Total cigarette consumption of residents and visitors (Top) - legal domestic sales and duty-free cigarettes imported personally (Down) | Demand data: N/A. Supply Data: Official revenue data. Legal import: Residents' and visitors' legal import of cigarettes was estimated using three alternative scenarios. | | The 2015 estimates were validated by a population survey that determined the proportion of local smokers purchasing illicit or imported cigarettes. | NA |
| Bui et al., 2022 | Malaysia | To measure the size of the illicit cigarette trade. | Increased penalties for the sale or purchase of illegal tobacco. | N/A | Consumption gap = Consumption – Legal Sales | Demand data: National Health and Morbidity Survey (1986, 1996, 2006, 2015, 2019) Global Adult Tobacco Survey 2011 Institute for Health Metrics and Evaluation data for adolescent smoking rate. Supply Data: Local production of cigarettes for the local market from the Ministry of Primary Industries, Malaysia Customs and import numbers from the Department of Statistics Malaysia | Billions of sticks | For years lacking national surveys, the Catmull and Rom (1974) interpolation method was utilised, adjusting underreporting within a 0–10% range. | No |
| Gallien & Occhiali, 2022 | Sierra Leone | To evaluate the effects of a tobacco tax increase in Sierra Leone on smuggling. | Introduced a new high excise tax on cigarettes. | N/A | Gap = Estimated consumption – legal production or imports | Demand data: District Health Survey for 2013 and 2019. Supply Data: Customs data and National Revenue Authority. | N/A | Assuming 20% underreporting, the authors estimated 40 models with varied assumptions. | No |
| Nguyen, M. T. et al., 2014 | Vietnam | To measure the magnitude of the illicit cigarette trade. | Changes in tobacco taxation and legislation on importing cigarettes. | N/A | Illicit cigarette consumption = Survey-based estimate – tax-paid cigarette sales | Demand data: Vietnam Living Standards Survey for 1998 and 2006, and Vietnam National Health Survey for 2002 Global Adult Tobacco Survey Vietnam for 2010 Supply Data: Tax-paid sales from the Ministry of Industry and Trade Trade data: United Nations' Comrade database | Millions of packs and Millions of dollars | Under-reporting at 10%, 20%, and 30% was applied for sensitivity analysis. | No |

*(Continued)*

**Table 4.** (Continued)

| Author, Year of publication | Country of origin | Aim/ purpose | Type of tobacco policies in the current study context | Potential policy scenario | Data analysis | Data source | Unit of measurement | Comment | Industry funded |
|---|---|---|---|---|---|---|---|---|---|
| Pava-nanunt, 2010 | Thailand | To investigate cigarette tax avoidance in Thailand and quantify its magnitude. To investigate government revenue loss from tax avoidance. | N/A | N/A | Estimated cigarette tax avoidance=Annual manufactured cigarette consumption - tax-paid sales Estimated tax avoidance=Sum of recorded exports to Thailand - recorded imports from other countries. | **Method 1:** <u>Demand data:</u> Cigarette consumption was estimated using the National Health and Welfare Surveys (1991, 1996, 1999, 2006) and Cigarette Smoking and Alcoholic Drinking Behaviour Surveys (2001, 2004). <u>Supply Data:</u> Cigarette production and sales data from the Excise Department and the Thailand Tobacco Monopoly. **Method 2:** <u>Trade data:</u> 16 years of recorded exports from all countries to Thailand obtained from the United Nations Commodity <u>Trade Statistical Data</u> (UN-Comtrade); recorded imports to Thailand retrieved from the Thailand Customs Department, Ministry of Finance. | Millions of Packs | Sensitivity analysis applied varying under-reporting percentages. Taxation, consumption, and smuggling were analysed in relation. | No |
| Nguyen, H. T. T. et al., 2020 | Vietnam | To examine the impact of higher tobacco taxes on illicit trade. | The tax share of the price was low despite a high special consumption tax on cigarettes. | Likely to change the tax structure and tax increases | Discrepancy = (domestic consumption - domestic production) + (legally exported - legally imported) If D>0, SMUGIN - SMUGOUT>0 =>SMUGIN>SMUGOUT; thus, Vietnam experienced a net illicit cigarette inflow. If D<0, SMUGIN - SMUGOUT<0 =>SMUGIN<SMUGOUT; thus, Vietnam experienced a net illicit cigarette outflow. | <u>Demand data:</u> Extracted from Global Adult Tobacco Surveys 2010 and 2015 to estimate cigarette consumption. <u>Supply Data:</u> Tax-paid sales data from Vietnam Tobacco Union, VINACOSH, Vinataba. <u>Trade data:</u> Cigarette import data from the Vietnam Tobacco Association and General Department of Customs. | Million Packs | Sensitivity analysis was conducted across four scenarios: No under-reporting 10% under-reporting 20% under-reporting 30% under-reporting | No |

*(Continued)*

**Table 4.** (Continued)

| Author, Year of publication | Country of origin | Aim/ purpose | Type of tobacco policies in the current study context | Potential policy scenario | Data analysis | Data source | Unit of measurement | Comment | Industry funded |
|---|---|---|---|---|---|---|---|---|---|
| Abola et al., 2014 | Philippines | To estimate the magnitude of illicit cigarette trade. | Weak tax administration. | The tax structure was simplified, leading to a single, uniform cigarette tax rate. | Discrepancy = Survey-based cigarette consumption estimates - cigarette removals (produced or licensed for sale, subject to excise tax and value added tax) Discrepancy = Recorded imports by the Philippines - exports to the Philippines recorded by the trading partner. | Demand data: Tobacco consumption was estimated from expenditures data using the Family Income and Expenditure Survey collected every three years by the National Statistics Office. Average price per brand obtained from a 2003 survey by the Bureau of Internal Revenue. Brand market share determined each brand's cost to produce a single average price, adjusted for each survey year using the cigarette consumer price index. Data on adolescent smoking was from the 2007 Global Youth Tobacco Survey and the 2009 Global Adult Tobacco Survey. Supply Data: Tax sales were derived from the Bureau of Internal Revenue. Trade data: United Nations Commodity Trade Database. | Billion packs | Sensitivity analysis covered under-reporting at 10%, 20%, and 30%, considering smoking by tourists and migrant workers. | No |
| Ahsan et al., 2014 | Indonesia | To measure the magnitude of illicit cigarette consumption. | There were a low tax rate and a complex tax system. | Potential increase in tax | Comparison: Survey-based tobacco consumption vs. legal sales<br><br>Trade discrepancy = exports recorded by the country of origin - imports recorded by Indonesia<br><br>Consultations with experts to understand cigarette origin. | Demand data: Estimated tobacco consumption derived from the National Socio-Economic Survey (1995, 2004), Household Health Survey (2001), Basic Health Research Survey (2007, 2010, 2013), and Global Adult Tobacco Survey 2011.<br><br>Supply Data: Based on excise stamp orders from the Directorate General of Excise and Customs.<br><br>Trade data: From the Central Board of Statistics.<br><br>The Directorate General of Customs was consulted. | Billions of sticks | Sensitivity analysis explored various underreporting assumptions. Estimated tax loss was calculated as the sum of estimated illicit domestic cigarettes multiplied by the average specific excise tariff and estimated smuggled cigarette volume multiplied by the excise tariff for imported cigarettes. | No |
| Guindon, G. E. et al., 2016 | Canada | To examine levels and trends in Canada of contraband cigarettes. | High tobacco tax. | N/A | Illicit tobacco = estimated consumption - tax-paid sales Analysed respondents' responses on smuggled cigarettes from individual-level surveys. | Demand data: The Canadian Community Health Survey and the Canadian Tobacco, Alcohol, and Drugs Survey (formerly Canadian Tobacco Use Monitoring Survey). Supply Data: Wholesale data reported by tobacco manufacturers to Health Canada. | N/A | Results for youth smokers were separately analysed using method 2. | No |

*(Continued)*

**Table 4.** (Continued)

| Author, Year of publication | Country of origin | Aim/purpose | Type of tobacco policies in the current study context | Potential policy scenario | Data analysis | Data source | Unit of measurement | Comment | Industry funded |
|---|---|---|---|---|---|---|---|---|---|
| Vellios, Nicole et al., 2022 | South Africa | To analyse different methods of measuring illicit tobacco trade. | Tax stamps and warning signs on cigarette packs are ineffective in identifying illicit tobacco. | N/A | Gap = Self-reported consumption estimates - legal sales Price threshold = Price point distinguishing legal from illegal cigarettes. The last purchase price was collected from primary and secondary data. | N/A | N/A | N/A | No |
| Kasri et al., 2021 | Indonesia | To estimate the volume and tax revenue loss due to the illicit cigarette trade | Complicated and weak tax system. | N/A | Gap = Consumption - Tax-paid sales Revenue loss = Illegal domestic cigarette production × (excise tax + value added tax + local cigarette tax) | Demand data: Ministry of Health reports and Statistics Indonesia. Supply Data: Ministry of Finance. | Billions of sticks | N/A | No |
| Paraje, Guillermo et al., 2023 | Chile | To assess the evolution of illicit tobacco trade | The most recent tobacco tax increase occurred in 2014. | N/A | Gap = total consumption – tax-paid sales | Demand data: National Drug Surveys in the General Population 2016, 2018 and 2020. Supply Data: Tax-paid sales data from Chilean Inland Revenue Service | Billions of sticks | N/A | No |
| Goodchild et al., 2020 | India | To estimate the magnitude of illicit cigarette consumption | N/A | N/A | Annual illicit cigarette consumption = annual illicit cigarette consumption - legal sales | Demand data: Global Adult Tobacco Surveys 2009–2010 and 2016–2017 Supply Data: calculated from annual revenue on cigarettes | Billions of sticks | N/A | No |

N/A = Not available. †Only the abstract was available.

conditions. The CGA method served to monitor the extent of the current and past illicit tobacco trade, which was ultimately useful to assess the trends of illicit tobacco trade over the past years, rather than estimating future trends in the illicit trade. CGA is grounded in a simple mathematical model: the estimated amount of illicit tobacco trade or consumption gap, representing the difference between the estimated amount of tobacco consumption and legal sales.

Nine of the 16 studies exclusively employed the CGA method [28–36]. Four other studies [37–40] incorporated the trade deficit method in their research and used both CGA and the trade deficit method. Trade deficit methods involved calculating tax avoidance by summing the reported volumes of tobacco exported from various countries to a specific country and cross-referencing them with the country's importation records. One study [41] used the price threshold method. This method involved applying a price threshold on the last purchased tobacco of participants to distinguish legal tobacco from illegal tobacco. Another study analysed survey responses regarding smuggled cigarettes [42] and one study [43] estimated revenue loss due to the trade in illicit tobacco.

CGA methods require two major sources of data: demand data and supply data. Demand or estimated consumption data were primarily calculated using national survey data (12 of the 16 studies, Table 3). Supply or legal sales data were predominantly obtained from the customs and revenue sector (10 of the 16 studies, Table 4) or other government agencies, such as the Ministry of Trade (n = 1), Ministry of Health (n = 1), or Ministry of Finance (n = 1) of the respective countries (Table 4). In one study (40), researchers utilised data from Euromonitor International because country-level data were unavailable [44]. The study acknowledged that Euromonitor International's illicit trade estimates were inconsistent; however, it relied on Euromonitor International's registered sales data for certain countries, assuming that there are fewer incentives to manipulate data on tobacco production or legal sales.

An important aspect of correctly estimating demand or consumption data is the consideration of under-reporting. Under-reporting of cigarette consumption can cause issues when using the CGA method. To compensate for the under-reporting, two of the 16 studies utilised an 'uplift factor' (Table 4) [28,35]. The 'uplift factor' was calculated using the formula of legitimate consumption divided by total consumption for a particular year. Researchers multiply the estimated consumption by the uplift factor to achieve the final estimated consumption, which is more than the estimated consumption [9].

As part of sensitivity analyses, eight studies also used the 'different percentages' method, which assumes under-reporting from 0% to 40% (Table 4) [32–35,38–40,45], while one study [30] employed both the 'uplift factor' and 'different percentages' methods.

The CGA method was used to estimate the amount of, or trends in, illicit tobacco at a national level; however, it was not applied to specific regions within a country or sub-populations due to the method's limitations and the limited availability of quality data. Therefore, equity aspects were not reflected in any CGA study.

## Non-CGA related studies

Twenty-three studies that applied other economic or econometric methods were reviewed. All 23 studies focused on demand: 22 on tobacco and one on illicit cannabis [46]. Nine studies examined the illicit tobacco trade [42,47–54], either past or existing demand, or the potential demand for certain illicit products in the event of future policy changes. A summary of the models, their purposes, strengths, and limitations is presented in Table 5.

The Hursh and Silberberg's exponentiated model was used in six of the 23 studies [48,49,55–58]. This behavioural economic model was used in these studies to analyse the price elasticity of various tobacco products, including illicit tobacco, by examining how changes in price, availability, and taxation policies influence consumer behaviour, product substitution, and demand shifts.

Five studies used discrete choice models [45,46,59–61]. Like the behavioural economic model, discrete models were used to assess the demand and preference for different tobacco products and marijuana and to test the preference for different smoking warning signs.

Another approach, the extended cost-benefit analysis, included the public health impacts and was holistic; it required extensive data and several assumptions. Static partial equilibrium was a focused model for tax and price simulation for cigarettes. In the case of tobacco, the scope was narrowed, and, with other potential market interactions apart from the illicit tobacco trade, such as substitution with other tobacco products, cross-border shopping, health care system implications, and macroeconomic effects, were not considered [62].

Acknowledging the strengths and weakness of different models, Tarantilis et al. applied three consumption models for comprehensive analysis of smoking behaviours: 1) conventional demand (simple to be applied, but ignores the role of addiction in driving behaviour, and time-dependent behaviours); 2) myopic addiction (accounts for addiction, but neglects the future potential consequences of current smoking behaviour, such as health deterioration or economic costs), and 3) rational addiction (considers both past and future consumption, but was based on the assumption that consumers behaved with perfect rationality) [63].

 

**Table 5. Summary of non-CGA-based economic and econometric analysis with the purposes, strengths, and limitations of the models.**

| Studies | Type of model | Purpose of the studies | Strengths | Limitations |
|---|---|---|---|---|
| Heckman et al., 2017; Denlinger-Apte et al., 2021; Freitas-Lemos et al., 2021; Tucker et al., 2018; Tucker, 2017 and Freitas-Lemos et al., 2023 | Hursh and Silberberg's Exponentiated Demand Model (Behavioural Economic Model) | To analyse the price elasticity of various tobacco products, including illicit tobacco, by examining how changes in price, availability, and taxation policies influence consumer behaviour, product substitution, and demand shifts | Captured behavioural economic responses; flexible for different products and prices. | Assumed consistent rationality; may require experimental data. |
| Buckell et al., 2017; Ning & Villas-Boas, 2019; Verbič et al., 2019; Tangtammaruk, 2017 and Guindon, G. Emmanuel et al., 2024 | Discrete Choice Models included Binary Choice Models (e.g., Binary Logit and Probit), Multinomial Logit Models, and Conditional Logit Models. Additionally, Advanced Logit Models such as Mixed Logit, Latent Class Logit, and Dynamic Logit) | to test the demand and preference for different tobacco products in various policy scenarios (for example, banning certain tobacco products), the preference for different smoking signs, and the demand for marijuana of people with different social and economic characteristics, and the demand for various tobacco labels and packaging. | Allowed rich preference modelling; suitable for policy simulations. | Data-intensive |
| Divino et al., 2022 | Extended Cost–Benefit Analysis and A Static Partial Equilibrium Model | To examine the impact of tax reform on cigarette prices, consumption, and tax revenue and estimate the size of the illicit tobacco market. | Captured Both fiscal and public health outcomes; allows policy simulation. | Complex data needs, the equilibrium model ignored broader market interactions. |
| Tarantilis et al., 2015 | Consumption Models: Convention Demand Model, the Myopic Addiction Model, and the Rational Addiction Model. | To evaluate smokers' sensitivity to cigarette prices and income changes and project the health benefits of future tax increases. | Progressively captured addiction behaviour; supported forecasting. | Assumption-heavy (primarily rational model); hard to validate empirically. |
| Ouellet et al., 2010 | Risk Prediction Models | To examine consumer behaviour toward cigarettes and illicit tobacco in response to tax decreases | Helpful in estimating policy risk effects; allowed probabilistic modelling. | Dependent on assumptions, illicit behaviour was challenging to validate. |
| Petruzzello, 2019 | A Forward-looking Behavioural Model based on Gruber and Köszegi's Framework | To introduce a new test for forward-looking behaviour | Incorporated time inconsistency and policy-relevant insights. | Complex parameter estimation assumed rational forecasting. |
| Tuchman, 2019 | Integrated Micro-Macro Model of Demand | To assess the impact of E-cigarette (EC) advertising on cigarette demand. | Combined individual and market data; models substitution effects. | Data-heavy; assumed rational response to ads. |
| DeCicca, 2013 | Endogenous Switching Regression Model | To conduct a welfare economics analysis of cigarette tax avoidance. | Captured decision to avoid taxes; informs optimal taxation. | Methodologically complex; needed rich individual-level data. |
| Jurušs et al., 2018 | Multiple and Non-Linear Regression Model | To estimate the impact of retail tobacco product display bans on smoking rates. | Models of complex, non-linear relationships allowed for control of confounders. | It required high-quality data; proving causality was complex. |
| Cobiac et al., 2015 | Dynamic Projection Model | To assess whether regular tax increases would be sufficient to achieve New Zealand' Smoke-free goal by 2025. | Precise policy forecasting; data-driven projections. | Assumed stable trends; may miss equity factors, although the model was applied for two ethnic groups (Māori and non-Māori) populations |

*(Continued)*

**Table 5.** (Continued)

| Studies | Type of model | Purpose of the studies | Strengths | Limitations |
|---|---|---|---|---|
| Irvine & Sims, 2012 | Demand-driven Analytical Model. | To analyse the impact of tax and price changes on the cigarette market composition | Captured market composition shifts; supported tax policy design. | The static model relied on illicit market assumptions. |
| Van Walbeek, 2014 | An Econometric Regression Model, specifically Time-Series and Panel Data Regression Techniques. | To evaluate cigarette excise tax revenue trends, comparing recent years to previous decades to determine if revenue has fallen short of budget projections and analyse illicit market size trends since 1995 | Strong time-based trend analysis; robust econometric technique. | Needed reliable illicit data; context-specific. |
| Stoklosa, 2020 | Econometric Modelling (Pooled Time-Series Data and Fixed-Effects Models) | To study the effects of EU Member States' cigarette price differences on cross-border purchases. | Controlled for fixed country traits; good for policy comparisons. | May miss informal trade; assumed rational behaviour. |
| Wang et al., 2019 | A Two-Way Fixed Effects Model | To analyse cigarette tax compliance using the first nationwide littered pack collection. | Novel data source; controlled for time and location effects. | Littered packs may not be representative; they assumed unchanging fixed effects. |

Risk prediction models were used to predict consumer behaviour under different tax policy scenarios, including illicit tobacco. However, such a model depended on several assumptions, such as a direct relationship between cigarette tax levels and smuggled cigarette consumption, minimal influence of tax changes on smoking initiation or cessation, and the adequacy of self-reported survey data as a proxy for illicit tobacco use because data on the illicit tobacco market were limited [51]. Similarly, a forward-looking behaviour model could help capture time-inconsistent (present-biased) preferences. However, the model involved complex parameter estimation and assumed rational decision-making, an assumption that may not be applicable to tobacco addiction [64].

The integrated micro–macro demand model developed by Tuchman (2019) incorporated addiction and persistence effects through a state-dependent framework, where past consumption increased the likelihood of current consumption. However, the model assumed rational behaviour and may omit unobserved influences on demand [65]. An endogenous switching regression model was useful for capturing tax avoidance behaviour and, therefore, could inform policy, including tax policy. However, it was methodologically complex and required detailed individual-level data on purchasing behaviour and tax avoidance. It also relied on several assumptions, such as the accuracy of self-reported cross-border purchases, the strong responsiveness of tax avoidance to price differentials, the rationality of consumer behaviour, and the absence of unobserved confounders once observable factors were accounted for [47].

Multiple and non-linear regression models captured complex, non-linear relationships and were used in policy evaluations on retail tobacco product display bans on smoking rates. The models required high-quality data, did not establish causality, and were sensitive to model specification [66]. Another model that could be used for policy evaluation was the dynamic projection model, which was used to evaluate whether regular tax increases would suffice for NZ to achieve its Smokefree 2025 goal. This model forecasted future smoking prevalence based on current trends and policy interventions. However, it oversimplified factors, assumed consistent trends [67].

A demand-driven analytical model was used to comprehensively analyse the impact of tax and price changes on the composition of the cigarette market, with particular focus on the shift between legal and illegal cigarette consumption [50]. The model's limitations included the inability to accurately measure illegal cigarette consumption, which relied on simplifying assumptions and static analysis.

**Table 6. Summary of the characteristics of studies using non-CGA-based economic or econometric methods.**

| Author, Year of publication | Country of origin | Aim/ Purpose | Type of tobacco policies in the current study context | Potential policy scenario | Data source/Data collection | Data analysis |
|---|---|---|---|---|---|---|
| Heckman et al., 2017 | Netherlands | To test the substitutability of nicotine replacement therapy (NRT), electronic cigarettes (EC), and very low nicotine content (VLNC) cigarettes in an online experimental tobacco marketplace (ETM). | VLNC cigarettes were never introduced in the Netherlands. | Novel products such as VLNC cigarettes were introduced. | An ETM was developed to simulate the decision-making of 840 participant smokers under four policy scenarios: • Conventional cigarettes were banned. • VLNC cigarettes were not available. • All products were available. • EC was not available. Participants made weekly purchases of conventional cigarettes at four random prices: one-half market price (MP), MP, 2x MP, and 4x MP. | Own price elasticity derived from exponential demand curve modelling: $\log_{10}Q = \log_{10}Q_0 + k\,(e - \propto Q0\ C - 1)$ The cross-price elasticity of other products was derived from the linear regression of the group means of the fixed-price product. |
| Denlinger-Apte et al., 2021 | United States of America (USA) | To explore menthol smokers' behaviours at varying prices of menthol cigarettes, with or without the availability of menthol little cigars and cigarillos (LCC) or menthol EC, using ETM. | Menthol cigarettes comprised over 30% of cigarette sales in the USA. | There was potential for introducing new regulations concerning menthol cigarettes. | Participants (n = 43) were asked to conduct hypothetical purchases of tobacco and nicotine products, including their usual brand of menthol and non-menthol cigarettes, menthol, mint, and tobacco-flavoured LCC, EC, snuff, snus, nicotine gum, and patches under various scenarios. | Demand curves were created for each product using the exponential demand equation, with cross-price elasticity estimates calculated to determine the degree of product substitution. |
| Freitas-Lemos et al., 2021 | USA | To experimentally study the impact of vaping and flavoured EC bans on the likelihood of buying illicit EC products and factors influencing purchases from a hypothetical illegal ETM. | Certain states and cities have banned access to EC products. | EC products or flavoured ones were banned. | An experiment with 150 participants, including exclusive cigarette smokers, EC users, and dual users, was conducted. They completed hypothetical purchases from online legal and illegal ETMs. | Marketplace preferences were estimated using mixed-effects logistic regression to assess differences between illegal and legal ETMs across various ban scenarios and cigarette prices. Reasons for choosing legal or illegal ETMs were compared to determine the percentage of agreement. Risk-taking data in the illegal market was analysed using an exponentiated function derived from the exponential demand equation. |
| Buckell et al., 2017 | USA | To offer policy-relevant estimates of flavour bans on combustible cigarettes and ECs and their impacts on demand for both types. | Over 7,000 EC flavours were available, with all flavours except menthol banned in combustible cigarettes. | The Food and Drug Administration of the USA had considered banning EC flavours and/ or menthol in combustible cigarettes. | A Discrete Choice Experiment (DCE) enlisted 2,031 adult smokers and recent quitters. Respondents chose their preferred cigarette type from four options, each described by attributes including flavour, health impact, nicotine amount, and price. | Multinomial logit models were used to analyse respondents' choices. |

*(Continued)*

**Table 6.** (Continued)

| Author, Year of publication | Country of origin | Aim/ Purpose | Type of tobacco policies in the current study context | Potential policy scenario | Data source/Data collection | Data analysis |
|---|---|---|---|---|---|---|
| Tucker et al., 2018 | NZ | To assess the price elasticity of regular cigarettes and VLNC cigarettes and to determine the cross-price elasticity of VLNC cigarettes, indicating the quantity of regular and VLNC cigarettes participants would purchase at different prices of VLNC cigarettes through a simulated cigarette purchase task (CPT). | Reducing nicotine content in cigarettes was not included in the policies. | There was a potential reduction in the nicotine content of cigarettes. | Data was collected through the experiment (simulated demand tasks) with 40 smokers. | Hursh and Silberg's exponentiated model determined the price elasticity of VLNC and regular cigarettes. Cross-price elasticities were calculated for individual participants by examining the regression slopes of (log) VLNC demand on (log) tobacco cigarette price. |
| Tucker, 2017 | NZ | To compare simulated demand for tobacco cigarettes, reactions to initial use of nicotine-containing electronic cigarettes, and the influence of nicotine-containing ECs availability on tobacco demand among NZ European and Māori/Pacific Island smokers. | NZ imposed a high tobacco tax, and nicotine-containing ECs were legally prohibited from sale. | Nicotine-containing ECs had the potential to complement existing tobacco control strategies by reducing demand for tobacco cigarettes. | The 210 participants were asked to complete CPT in the experiment and were allowed to sample nicotine-containing ECs. 30.1% identified as Māori/Pacific ethnicity and 69.9% as NZ European and other ethnicities. | Several analyses were conducted to characterize demand curves, including fitting the exponentiated version of Hursh and Silberg's demand model proposed by Koffarnus et al. (2015). |
| Tangtammaruk, 2017 | Thailand | To test preferences regarding smoking warning signs among smokers and non-smokers, and to evaluate the factors influencing the prevalence of youth smoking. | The standard no-smoking or prohibition sign featured a red circle with a red diagonal line through a picture of a cigarette. It had served as a smoking prevention tool in Thailand's schools, universities, and public places since 1992. | NA | Revealed preferences for smoking warning signs were collected by observing the choices of 860 individuals, both non-smokers and smokers, in real-life situations. Stated preferences were gathered by presenting five types of smoking warning signs (1) the standard no smoking sign, signs with the application of (2) Loss Aversion (public health warning signs), (3) Altruism (signs with the picture of second-hand smokers such as pregnant mother, child, elder), (4) Herd Behaviour (signs with celebrities promoting no smoking), and (5) Information (signs with information about the negative effect of smoking) are added in the model. | A descriptive analysis was conducted to compare revealed and stated preferences. An econometric binary choice model was utilised to examine characteristics affecting smoking prevalence. |

*(Continued)*

**Table 6.** (Continued)

| Author, Year of publication | Country of origin | Aim/ Purpose | Type of tobacco policies in the current study context | Potential policy scenario | Data source/Data collection | Data analysis |
|---|---|---|---|---|---|---|
| Verbič et al., 2019 | Slovenia | To analyse marijuana demand and evaluate the potential size of the industry by estimating consumption levels and retail expenditure. | There were relatively mild criminal sanctions for marijuana possession. | The legalisation of marijuana was being considered. | Data from the 2011 and 2012 national surveys, which included 7,200 individuals in 2011 and 8,000 individuals aged between 15 and 64 in 2012, were utilised. An online survey comprising 730 observations was conducted among national marijuana users in 2015. | Marijuana demand was analysed by estimating two univariate discrete choice models, linking the probability of marijuana participation to individual socio-economic characteristics, using micro-unit data from the national survey. Data from the online survey was utilised to assess the marijuana market. |
| Divino et al., 2022 | Brazil | To investigate the effects of tax reform on cigarette prices, consumption, and tax collection, and to estimate the size of the illicit cigarette market. | Existing tobacco control measures, including taxation, were in place. However, the tax system was highly complex, with consumption taxes being particularly convoluted. | The potential tax reform included simplifying the tax scheme. | Cigarette consumption patterns were derived from four surveys: National Household Sample Survey 2008, National Health Survey 2013, National Health Survey 2019, Surveillance of Risk Factors and Protection for Chronic Diseases by Telephone Survey (Vigitel) 2018 and 2019 | An Extended Cost-benefit analysis framework was used to assess the impact of tobacco taxation in Brazil. This approach was beyond traditional cost-benefit analyses by incorporating both tax policies' direct and indirect effects on public health and economic outcomes. Additionally, the researchers utilised a static partial equilibrium model to simulate the effects of tax reforms on the cigarette market and associated tax revenues. |
| Tarantilis et al., 2015 | Greece | To assess smokers' sensitivity to cigarette prices and changes in consumer income and project health benefits of further tax increases. | In 2010, tax restructuring accounted for 86% of cigarette retail prices; however, by 2012, the sales tax was fixed at 23% of the retail price. | Potential tobacco tax increase. | Data on annual cigarette consumption from 1994 to 2012 were obtained from the Greek Ministry of Finance, along with weighted average cigarette prices. Gross Domestic Product data was sourced from the World Bank. The National Action Plan for Cancer (2008), an anti-smoking campaign, and regulations for smoking bans and restrictions introduced in 2002 (amended in 2010) were included as dummy variables. | Three models were used for data analysis of cigarette consumption: the Convention Demand Model, the Myopic Addiction Model, and the Rational Addiction Model. Independent variables included Weighted Average Price as a proxy for cigarette price, Gross Domestic Product as a proxy for consumers' income, and dummy variables reflecting smoking restrictions and anti-smoking campaigns. Four scenarios of tax increases were calculated. |
| Ouellet et al., 2010 | Canada | To explore consumer behaviour towards cigarettes and illicit products as taxes decrease, while considering individual factors influencing these behaviours. | In 1994, the Canadian Federal Government reduced excise taxes from $10.35 to $5.36 per carton to curb cross-border smuggling. By 2000, tax levels had reverted to pre-1994 rates. | Potential cigarette tax reduction. | A 1-year longitudinal study was conducted among a nationally representative sample of Canadians aged 15 and older in four cycles. Respondents reported their smoking status before and after tax cuts in three subsequent periods. The first cycle surveyed 15,804 Canadians, focusing on younger and lower-income groups. By the fourth cycle, 4,685 respondents were lost, leaving 11,119 individuals who responded to all four cycles. | Tax influences on quitting and cigarette smuggling were estimated using risk prediction models. |

*(Continued)*

| Author, Year of publication | Country of origin | Aim/ Purpose | Type of tobacco policies in the current study context | Potential policy scenario | Data source/Data collection | Data analysis |
|---|---|---|---|---|---|---|
| Petruzzello, 2019 | USA | To introduce a new test for forward-looking behavior based on implementing smoking restrictions. | Smoking restrictions in public places, including bars, restaurants, and workplaces, were announced and established between 2006 and 2010, alongside the increased State cigarette excise tax. | N/A | The Nielsen Homescan Data Set included a national panel of 26,630 households that recorded cigarette purchases from 2006 to 2010. Data on state cigarette excise tax increases from 2006 to 2010 were obtained from the Federation of American Tax Administrators, while information on smoking restrictions was sourced from the Centres for Disease Control and Prevention. | A model like equation (1) of Gruber and Köszegi was developed and estimated to examine forward-looking behaviours regarding household cigarette purchases. |
| Tuchman, 2019 | USA | To assess the impact of ECs advertising on cigarette demand. | While TV advertising for traditional cigarettes had been banned in the USA since 1971, advertising for ECs remains unregulated. | Lack of ECs advertisements. | The study utilised retail sales data recorded weekly in the Nielsen Database from 2010 to 2015, covering 455 tobacco cigarette brands and 9,257 unique Universal Product Codes (UPCs). Household purchase data included daily UPC-level purchases for approximately 50,000 US households between 2010 and 2015, with 2,288 households making 10,962 purchases of any ECs during this period. Advertising data consisted of weekly product-level television advertising data from 2009 to 2015 sourced from Nielsen. | The study developed the Integrated Micro-Macro Model of Demand and applied the estimated preference parameters to predict the counterfactual cigarette demand without EC advertising. |
| Ning & Villas-Boas, 2019 | USA | To Investigate the effects of changes in label informativeness on consumer choice in Chapter 3. | The Tobacco Control Act, introduced in 2009, banned the usage of strength descriptions such as "regular," "light," or "ultra-light" on any marketing or packaging materials. Tobacco companies were only permitted to differentiate their strengths through colour codes. | There were potential modifications to the labels of tobacco products. | Nielsen's retailer scanner and household panel data from 2007 to 2012 were utilised. The Nielsen database included retail scanner data from approximately 35,000 grocery, drug, and other stores, following a panel of around 60,000 U.S. household purchases. | Discrete choice models were fitted, including the estimation of a simple logic model, to account for preference heterogeneity, state dependence, and price endogeneity, with the aim of understanding the change in price sensitivity. |
| DeCicca, 2013 | USA | To conduct a welfare economics analysis of cigarette tax avoidance. | States imposed varying excise tax rates on cigarettes. | Federal excise tax on cigarettes. | Cross-border cigarette purchase data from the 2003 and 2006–2007 cycles of Tobacco Use Supplements to the USA Current Population Survey. | Estimated an Endogenous Switching Regression model for border crossing and cigarette prices. Developed an extension of the standard formula for optimal Pigouvian corrective tax to account for consumer tax avoidance through purchases in nearby lower-tax jurisdictions. Conducted illustrative calculations for optimal taxes and sensitivity analyses based on various assumed parameters. |

*(Continued)*

**Table 6.** (Continued)

| Author, Year of publication | Country of origin | Aim/ Purpose | Type of tobacco policies in the current study context | Potential policy scenario | Data source/Data collection | Data analysis |
|---|---|---|---|---|---|---|
| Jurušs et al., 2018 | Latvia | To estimate the impact of retail tobacco product display bans on smoking rates. | A retail display ban was implemented in Latvia. . | N/A | Prices and retail volumes were sourced from statistical data. For the price forecasting model from 2018 to 2020, excise tax rates were obtained from the law "On excise tax." Data sources included the number of consumers from 2006 to 2016 from the Centre for Disease Prevention and Control, Latvia(SPKC, 2012, 2014, 2016); information on citizen numbers and changes from 2006 to 2017 from Central Statistical Bureau of Latvia (CSP); average income of the population from statistics from 2006 to 2016 from CSP 2017, and Ministry of Finance forecasts for future years. A retailer survey with 59 participants was conducted. | The multiple regression method was employed, and cigarette prices, excise tax, consumer income, the number of smokers, the illegal market, and the tobacco display ban were identified as critical factors in cigarette consumption. A non-linear regression model was developed to forecast legal consumption and assess income. |
| Cobiac et al., 2015 | NZ | To assess whether regular tax increases would be sufficient to achieve NZ's smoke-free goal by 2025. | NZ aimed to achieve a smoke-free status (with less than 5% smoking prevalence) by 2025. Tobacco control policies included Quitline, cessation support, and a commitment from 2011 to 2016 to implement 10% annual increases in tobacco excise tax. | N/A | Data on cigarette prices (181 product varieties) from the online supermarket Countdown were collected. Illicit product prices were calculated using three scenarios: population data from Statistics NZ, mortality data from the NZ Census mortality study, and smoking prevalence and intensity data from the NZ Health Survey. | The dynamic projection model simulated smoking prevalence from 2011 onward with annual excise increases of 0%, 5%, 10%, 15%, and 20%. Annual cigarette prices were calculated until 2060, considering changes in excise amounts, goods and services tax, tax pass-through, and illicit market activity. The modelling comprised two stages: first, determining current probabilities of smoking uptake and cessation; second, simulating future smoking prevalence using a no-tax-increase scenario and annual tax increases, using generated probabilities and forecasted populations, mortality, and smoker mortality risk trends. |
| Irvine & Sims, 2012 | Canada | To analyse the impact of tax and price changes on the cigarette market composition using a demand-driven analytical model. | There were claims to cut tobacco taxes to curb illicit trade. | Tax reduction. | N/A | The study modelled consumer reactions to price gaps between legal and illegal products, assessing the impacts of tax cuts on consumption, government revenue, and illegal market size. Four scenarios were devised: Tax reduction on legal products Strengthened enforcement against illegal products Combined policy approach Tax mix adjustment It tracked changes in % spent on cigarettes, price index, legal vs. illegal market share, tax revenue, and tax revenue per $ spent on cigarettes. |

*(Continued)*

**Table 6.** (Continued)

| Author, Year of publication | Country of origin | Aim/ Purpose | Type of tobacco policies in the current study context | Potential policy scenario | Data source/Data collection | Data analysis |
|---|---|---|---|---|---|---|
| Van Walbeek, 2014 | South Africa | (1) Evaluate cigarette excise tax revenue trends, comparing recent years to previous decades to determine if revenue has fallen short of budget projections. (2) Analyse illicit market size trends since 1995. | There was a stable tobacco policy, yet the excise tax structure has been unchanged since 2004. | N/A | Excise revenue data for beer, spirits, cigarettes, and cigarette tobacco were sourced from Auditor-General reports, Treasury's Budget Review, and GDP data from the African Reserve Bank. Actual cigarette prices were calculated using the tobacco price index divided by the Consumer Price Index from Statistics South Africa's monthly publication P0141. | Used an econometric regression model, specifically time-series and panel data regression techniques. Mean and root mean squared percentage errors were computed to assess forecast accuracy. This analysis included cigarettes, beer, and spirits for comparison. A simulated analysis of budgeted versus actual quantities was performed to estimate the illicit market share. |
| Stoklosa, 2020 | European Union (EU) | To study the effects of EU Member States' cigarette price differences on cross-border purchases. | Despite the EU's minimum tax requirements in the tobacco tax directive, significant price variations among member states persisted. | Tax harmonization. | Data sources were cigarette consumption and prices, border population, and gasoline prices from the European Commission; cigarette prices for EU neighbouring countries from Euromonitor; GDP, inflation, and population data from the World Bank; and Non-price tobacco control scores from WHO. | Pooled time-series data and fixed-effects models to analyse how cross-border price differences influence cigarette purchases. |
| Wang et al., 2019 | USA | To analyse cigarette tax compliance using the first nationwide littered pack collection. | Tobacco tax rates varied among states. | N/A | Littered cigarette packs were collected from 160 nationwide communities, representing enrolment areas for 8th, 10th, and 12th-grade public school students in the continental US between May and July 2012. | A two-way fixed effects model was used. Evidence of cigarette tax noncompliance was integrated with data on potential determinants to estimate parameters and constructed incentives for noncompliance. |
| Freitas-Lemos et al., 2023 | USA | To estimate the impact of banning menthol cigarettes and decreasing allowable cigarette filter ventilation levels on demand for illegal cigarettes | The menthol cigarettes sales were banned at the State level | The potential banning of menthol cigarettes and filter ventilation at the country level | Legal and illegal ETMs were developed in two experiments to collect participants' responses on illegal marketplaces. Experiment 1 was conducted with 128 participants on different scenarios relating to the menthol cigarette ban. Experiment 2 was conducted with 226 participants on different scenarios regarding the high-ventilation ban. Participants were randomly assigned to one of four groups in two experiments, based on a 2×2 table design involving product standard and illegal marketplace availability as independent variables. In Experiment 1, groups were: 1) menthol cigarettes banned without illegal option, 2) menthol cigarettes banned with illegal option, 3) menthol cigarettes not banned without illegal option, and 4) menthol cigarettes not banned with illegal option. In Experiment 2, groups were: 1) high-ventilated cigarettes banned without illegal option, 2) high-ventilated cigarettes banned with illegal option, 3) high-ventilated cigarettes not banned without illegal option, and 4) high-ventilated cigarettes not banned with illegal option. | A linear mixed-effects model was employed to assess the interaction between menthol availability and tobacco product category, incorporating a participant random effect on the total budget spent per product category. The model with the lowest Bayesian Information Criterion (BIC) was selected as optimal, and its covariates were retained for subsequent analyses. |

*(Continued)*

**Table 6.** (Continued)

| Author, Year of publication | Country of origin | Aim/ Purpose | Type of tobacco policies in the current study context | Potential policy scenario | Data source/Data collection | Data analysis |
|---|---|---|---|---|---|---|
| Guindon, G. Emmanuel et al., 2024 | Canada | To examine the effects of plain and standardized packaging, warning on cigarettes, price, and the availability of illicit cigarettes on intention to purchase and risk perceptions | In November 2019, Canada became the country with the most comprehensive cigarette packaging regulations and in June 2022, Canada proposed to print health warnings on individual cigarettes, the regulations came into force on August 1, 2023, and are being implemented through a stepwise approach | N/A | A discrete choice experiment was conducted to capture the participants' responses on intention to purchase and risk outcomes. | Experimental choices were analysed within the framework of random utility theory, utilizing the conditional logit method. |

N/A = Not available

Econometric regression models (time-series and panel data regression techniques) provided comprehensive trend analysis, informed policy effectiveness, and assessed trends in the illicit market [53]. However, the model was limited by data accuracy, especially regarding the illicit cigarette market, and by static assumptions underlying the data-generating processes(58).Econometric modelling developed by Stoklosa (2020), using pooled time-series data and fixed-effects models, integrated temporal and cross-sectional data, controlling for unobserved heterogeneity, and provided policy-relevant insights for cross-border cigarette purchasing in the European Union countries. However, the model did not fully capture cross-border purchases, based on the assumptions of rational behaviour, and had limited external validity [52]. The two-way fixed effects model was used to analyse cigarette tax compliance using data from the first nationwide littered cigarette pack collection in the USA, and it informed the nationwide scope and policy on tax compliance. Since the model was based on discarded pack survey data (i.e., single time-point cross-sectional survey data), the packs may not represent the consumption patterns of the population, and the model assumed constant unobserved factors [54].

Finally, data were collected through a wide range of sources, including experiments (n = 9), surveys (including discarded pack surveys, n = 6), sales databases (n = 6), and various data sources, such as budgeted and actual revenue, prices, tax rates, and littered cigarette packs (n = 2). Table 6 provides details on the studies' characteristics, including their objectives, scenarios, and data sources.

## Critical appraisal of the studies from health equity perspectives

Given their shared methodology and national-level geographical coverage, the CGA studies were collectively considered a single study for critique. Among the 23 studies using economic/econometric models, most (n = 17, 74%) also performed policy scenario analyses. All studies conducted demand analysis, with most (n = 17, 74%) providing supplementary materials. Only six (26%) of the 23 studies investigated the equity implications of the assessed policies (Table 7).

**Table 7. Critical appraisal of strengths and limitations of the economic or econometric models from the health equity perspective.**

| Study | Main model | Population | Baseline scenario | Policy scenario | Demand analysis | Trans-par-ency | Equity |
|---|---|---|---|---|---|---|---|
| All 16 studies using the CGA method | Estimated illicit consumption = total estimated consumption – legal sales | No (The studies assessed the illicit market size for the whole nation) | Yes (assessed the existing illicit tobacco market size and illicit market size in the past, which provided a trend over the years) | No (The method could be used to estimate the illicit market size in the present and the past, regardless of the policy) | No | Yes | No Since it mainly estimated the illicit market at the national level. |
| Heckman et al., 2017 | Hursh and Silberberg's Exponentiated Demand Model (Behavioural Economic Model) | Yes | Yes | Yes | Yes | Yes | No |
| Denlinger-Apte et al., 2021 | | Yes | Yes | Yes | Yes | Yes | No |
| Freitas-Lemos et al., 2021 | | Yes | Yes | Yes | Yes | Yes | No |
| Tucker et al., 2018 | | Yes | Yes | Yes | Yes | Yes | No |
| Tucker, 2017 | | Yes | Yes | Yes | Yes | Yes | Yes (Ethnicity) |
| Freitas-Lemos et al., 2023 | | Yes | Yes | Yes | Yes | Yes | No |
| Tangtam-maruk, 2017 | Discrete Choice Models include Binary Choice Models (e.g., Binary Logit and Probit), Multinomial Logit Models, and Conditional Logit Models. Additionally, Advanced Logit Models such as Mixed Logit, Latent Class Logit, and Dynamic Logit) | Yes | Yes | No | Yes | No | Yes (Gender) |
| Ning & Villas-Boas, 2019 | | Yes | Yes | Yes | Yes | Yes | No |
| Guindon, G. E. et al., 2016 | | Yes | Yes | Yes | Yes | Yes | No |
| Verbič et al., 2019 | | Yes | Yes | Yes | Yes | No | Yes (Socio-economic characteris-tics) |
| Buckell et al., 2017 | | Yes | Yes | Yes | Yes | Yes | Yes (Adult smok-ers and recent quitters) |
| Divino et al., 2022 | Extended Cost–Benefit Analysis and A Static Partial Equilibrium Model | Yes | Yes | Yes | Yes | Yes | No |
| Tarantilis et al., 2015 | Consumption Models: Convention Demand Model, the Myopic Addiction Model, and the Rational Addiction Model. | Yes | Yes | Yes | Yes | Yes | No |

*(Continued)*

**Table 7.** (Continued)

| Study | Main model | Population | Baseline scenario | Policy scenario | Demand analysis | Trans-par-ency | Equity |
|---|---|---|---|---|---|---|---|
| Ouellet et al., 2010 | Risk Prediction Models | Yes | Yes | Yes | Yes | No | Yes (Smokers vs non-Smokers) |
| Petruz-zello, 2019 | A Forward-looking Behavioural Model, based on Gruber and Köszegi's Framework | Yes | Yes | No | Yes | Yes | No |
| Tuchman, 2019 | Integrated Micro-Macro Model of Demand | Yes | Yes | Yes | Yes | Yes | No |
| DeCicca, 2013 | Endogenous Switching Regression Model | Yes | Yes | Yes | Yes | Yes | No |
| Jurušs et al., 2018 | Multiple and Non-Linear Regres-sion Model | Yes | Yes | No | Yes | No | No |
| Cobiac et al., 2015 | Dynamic Projection Model | Yes | Yes | No | Yes | Yes | Yes (Ethnicity) |
| Irvine & Sims, 2012 | Demand-driven Analytical Model. | Yes | Yes | Yes | Yes | No | No |
| Van Walbeek, 2014 | An Econometric Regression Model, specifically Time-Series and Panel Data Regression Techniques. | Yes | Yes | No | Yes | No | No |
| Stoklosa, 2020 | Econometric Modelling (Pooled Time-Series Data and Fixed-Effects Models) | Yes | Yes | Yes | Yes | Yes | No |
| Wang et al., 2019 | A Two-Way Fixed Effects Model | Yes | Yes | No | Yes | Yes | No |

## Discussion

### Overview of economic and econometric methods used

This review found that current literature about estimating the size of the illicit tobacco trade used one of two perspectives: 1) estimating the amount and trends of illicit tobacco trade using the CGA method, or 2) estimating or predicting illicit tobacco use using non-CGA-based economic models and econometric analyses to estimate the demand.

### Strengths, limitations, and equity considerations of identified methods

The CGA method has been utilised to estimate both the current extent of illicit trade and historical trends, depending on the availability of data. However, the method has several limitations. It cannot be used to estimate future illicit tobacco trade. Although the method can generate state- or subpopulation-level estimates when sufficient data exist, limited data availability has resulted in its predominant use at the national level. These limitations mean that data for specific popula-tion subgroups are not available, which limits the method's applicability whenconsidering health equity. One possible rea-son for this limitation is the nature of the demand and supply data, which are typically available only at the national level. In contrast, behavioural economic models (like Hursh and Silberberg's exponentiated model) are used to analyse the price elasticity of various tobacco products, including illicit tobacco, to test whether changes in price, product availability, and taxation policies can significantly influence consumer behaviour, including patterns of product substitution and shifts in demand for illicit tobacco. Such models can be applied to analyse the past or existing demand for illicit tobacco and predict the future demand for illicit tobacco on policy changes. Discrete choice models could also be used to assess the

past or existing demand for illicit tobacco and predict future demand. Including examining consumer preferences regarding various tobacco control policies [45,46,59–61]. Both behavioural economic and discrete choice models can be used to conduct sub-population analyses for priority populations; however, only a limited number of studies have taken such an equity perspective.

Cost-benefit analysis and partial equilibrium models [62] were useful for simulating tax policy changes' fiscal and consumption effects, including estimates of the illicit market share. However, since they were generally not designed to provide estimates for specific population subgroups, health equity perspectives cannot be applied.

Consumption Models such as the Rational Addiction Model incorporated behavioural responses to price and income changes, helping to project shifts in legal and illicit consumption. The models could be adapted to estimate the responses of specific populations if the disaggregated data were available. Risk Prediction Models [51]and Forward-looking Behavioural Models [64]captured consumer responses to changing tax environments, including transitions to illicit products. These models could be applied to specific populations, provided relevant individual-level data were available.

Econometric Regression Models, including Time-Series, Panel Data, and Fixed Effects Models [52–54] provided statistical frameworks for evaluating long-term trends, cross-border effects, and tax compliance, which were indirect indicators of the illicit tobacco trade. These models effectively analysed trends over time and across regions or demographic groups; however, the data for specific populations were limited due to the availability and quality of data at the population-specific level.

An integrated micro-macro model of demand [65] was used to assess product substitution and could be applied to estimate the substitution between legal tobacco and illicit products. Such a model could be applied to specific populations, given that the model used both market and individual-level data. However, the models were data-intensive, and the availability of representative data for specific populations is often limited.

Switching regression models [47] and non-linear regression approaches [66] could identify the determinants and outcomes of tax avoidance behaviours, including the decision to engage in illicit purchasing. Such models could assess illicit purchasing behaviour among specific populations if disaggregated individual-level data were available.

Dynamic projection models [67] were employed for policy simulations, including investigating long-term goals (e.g., achieving national smoke-free targets). By integrating assumptions about the prevalence and growth of illicit trade, these models could estimate the extent to which illicit consumption may undermine progress toward smoke-free targets or reduce the effectiveness of taxation policies. Such models may also be used to assess smoke-free targets for disadvantaged or high-risk groups.

## Implications for research, practice, and methodological advancement for illicit tobacco

In NZ, Phyo and Bullen (2025) applied CGA to assess the proportion of illicit tobacco in overall tobacco use, from 2012 to 2023 [6]. The trade comparison method could also be applied to this research to validate the estimates of the illicit tobacco market in NZ. If tobacco import and export data (from the concerned countries) to NZ are available from national and international data sources, the trade comparison method could be applied in the NZ context. However, since NZ's smoking prevalence is very inequitable among different socioeconomic groups [18],it is crucial to supplement this method with other methods to estimate the illicit tobacco trade more accurately in NZ. Furthermore, NZ's tobacco control policies are evolving in response to political efforts to achieve smoke-free goals [68]. Therefore, it is essential to anticipate potential changes in illicit tobacco use while accounting for forthcoming tobacco control measures.

Non-CGA-based economic or econometric models that assessed or predicted tobacco users' behaviour in engaging with illicit tobacco could be very useful for filling the gap in estimating future illicit tobacco use among all tobacco users and specific populations in NZ. Behavioural economic models, discrete choice models, consumption models, risk predictions, and forward-looking behavioural models could be used to estimate behaviours related to illicit tobacco use.

Econometric analyses to quantify tax avoidance behaviour could also be applied to the NZ context. However, cross-border shopping may not always be applicable in island nations like NZ.

Researchers collected data to assess future behaviours through experiments, such as tobacco market experiments, discrete-choice experiments, and simulated cigarette purchase tasks. Similar experiments or data collection under hypothetical policy scenarios could be applied to the NZ context, given that the Smokefree Environments and Regulated Products Amendment Regulations Act 2023 (to mandate a very low nicotine content standard for all tobacco, retailer reduction, and a smoke-free generation) has been recently repealed [69].

The behavioural economic models were mainly driven by product availability and price. Two such studies were conducted in NZ [57,58]. However, the studies did not assess potential illicit tobacco use. A similar experiment from a public health perspective using other potential factors influencing the behaviour of people who smoke, e.g., product availability and health knowledge, would be useful to investigate illicit tobacco use.

Overall, this review highlights that most economic models used to assess illicit tobacco trade have applied limited equity dimensions. The six studies that included subpopulation analyses conducted limited subgroup analyses based on selected sociodemographic and smoking-related characteristics, including age, sex, ethnicity, income, and smoking status. To strengthen future applications, the studies that applied economic or econometric models should incorporate disaggregated datasets, subgroup analyses, and equity-weighted parameters to better reflect the distributional impacts of tobacco control policies across vulnerable or priority populations. Integrating these approaches would enhance the capacity of existing models to assess not only overall effectiveness but also the fairness and inclusiveness of policy outcomes, thereby advancing a health equity perspective in tobacco control research.

The inclusion criteria for the review enabled identification of studies examining other forms of illicit trade, such as illicit alcohol or drugs, while the PCC framework focused specifically on active tobacco users. The use of broader inclusion criteria was intentional and enabled the identification of methodological approaches that could be transferable to illicit tobacco research. However, because the primary focus of this review was illicit tobacco, the search strategy excluded studies centred solely on illicit drugs. Although these additional studies primarily contributed to methodological insights rather than direct evidence on illicit tobacco markets, they were valuable in identifying adaptable analytical techniques— such as behavioural and econometric models—that could be refined and applied within illicit tobacco contexts.

## Strengths and limitations

This review clarified the best available options for measuring the illicit tobacco trade from a market perspective and outlined the strengths and limitations of specific economic or econometric methods used for assessing current and potential behaviours related to the demand for illicit tobacco from an equity perspective. Strengths of the review included: 1) the broad literature search (that included grey literature), helping to ensure the review was comprehensive; 2) assessment of the application of economic and econometric methods assessing illicit tobacco use from a health equity perspective, and 3) our selection criteria meant studies on economic and econometric methods applied not only to illicit tobacco but also to other tobacco and nicotine products, and in one case, to other illicit goods such as marijuana. Several limitations should be acknowledged. First, no assessment of the individual models employed was undertaken. Further, due to resource limitations, only studies published in English were included. However, we extensively searched the key public health and economics databases, as well as added expert-suggested articles and grey literature. By doing this, we are confident there is minimal selection bias. Future reviews could address language bias by engaging multilingual collaborators or by targeted searches for key non-English studies.

## Conclusion

Estimating the extent of the illicit tobacco trade presents challenges, not only because of its illicit nature but also because of the limitations of current methods for measuring it (and the often-limited availability of high-quality data).

The choice of the most suitable economic and econometric methods for measuring the illicit tobacco trade is ultimately dependent on the research question. If the aim is to understand the current national illicit tobacco trade from the market side (and past trends), a CGA is most appropriate. However, CGA is not suitable for assessing illicit tobacco use in specific subpopulations due to limited data availability and for predicting the impact of future policy scenarios. If the aim is to assess the past, existing, and potential behaviours of tobacco users in engaging with the illicit tobacco market, economic models (e.g., behavioural economics, discrete choice, consumption, risk prediction, and forward-looking models) are best utilised, especially if a health equity perspective is required. Given the above, triangulation of data (i.e., CGA and other non-CGA-based economic or econometric models, alongside surveys of people who smoke tobacco and discarded pack surveys) remains the best pathway forward. However, consideration of health equity perspectives is essential.

## Supporting information

**S1 File. Review protocol.** Full review protocol describing the objectives, inclusion criteria, search strategy, and methods used for data charting and synthesis, developed in accordance with the Joanna Briggs Institute framework for scoping reviews.
(DOCX)

**S2 File. PRISMA-ScR Checklist.** Completed Preferred Reporting Items for Systematic Reviews and Meta-Analyses extension for Scoping Reviews (PRISMA-ScR) checklist outlining the reporting items addressed in this review.
(DOCX)

## Author contributions

**Conceptualization:** Pyi Pyi Phyo, Natalie Walker, Braden Te Ao, Erwann Sbai, Chris Bullen.

**Data curation:** Pyi Pyi Phyo.

**Formal analysis:** Pyi Pyi Phyo.

**Investigation:** Pyi Pyi Phyo.

**Methodology:** Pyi Pyi Phyo, Natalie Walker, Braden Te Ao, Erwann Sbai, Chris Bullen.

**Project administration:** Pyi Pyi Phyo.

**Software:** Pyi Pyi Phyo, Chris Bullen.

**Supervision:** Natalie Walker, Braden Te Ao, Erwann Sbai, Chris Bullen.

**Validation:** Pyi Pyi Phyo.

**Visualization:** Pyi Pyi Phyo.

**Writing – original draft:** Pyi Pyi Phyo.

**Writing – review & editing:** Pyi Pyi Phyo, Natalie Walker, Braden Te Ao, Erwann Sbai, Chris Bullen.

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
