## [Decision Letter · Decision Letter 0]

10 Mar 2025

PGPH-D-24-02979

Economic and econometric methods to measure the illicit tobacco trade: a scoping review.

Dear Dr. Phyo,

Thank you for submitting your manuscript to PLOS Global Public Health. After careful consideration, we feel that it has merit but does not fully meet PLOS Global Public Health’s publication criteria as it currently stands. Therefore, we invite you to submit a revised version of the manuscript that addresses the points raised during the review process.

We look forward to receiving your revised manuscript.

Kind regards,

Chandrashekhar T. Sreeramareddy

Academic Editor

Journal Requirements:

Please send a completed 'Competing Interests' statement, including any COIs declared by your co-authors. If you have no competing interests to declare, please state "The authors have declared that no competing interests exist". Otherwise please declare all competing interests beginning with the statement "I have read the journal's policy and the authors of this manuscript have the following competing interests:"

Please amend your detailed Financial Disclosure statement. This is published with the article. It must therefore be completed in full sentences and contain the exact wording you wish to be published.If you did not receive any funding for this study, please simply state: “The authors received no specific funding for this work.”

Please provide separate figure files in .tif or .eps format.For more information about figure files please see our guidelines:https://journals.plos.org/globalpublichealth/s/figureshttps://journals.plos.org/globalpublichealth/s/figures#loc-file-requirements

Additional Editor Comments (if provided):

The two reviewers have raised several concerns though they recommended minor and major revisions, irrespective the manuscript has good potential to guide on methods to be used for assessing the extent of illicit tobacco trade in NewZealand and beyond. I noticed that the opposite of inclusion are written as exclusion criteria. Could you remove exclusion criteria (these need not necessary to be there). If you had to exclude any articles among those included after title, abstract or full text review, please state the reasons.

Reviewers' comments:

Reviewer's Responses to Questions

**Comments to the Author**

1. Does this manuscript meet PLOS Global Public Health’s publication criteria ? Is the manuscript technically sound, and do the data support the conclusions? The manuscript must describe methodologically and ethically rigorous research with conclusions that are appropriately drawn based on the data presented.

Reviewer #1: Yes

Reviewer #2: Partly

2. Has the statistical analysis been performed appropriately and rigorously?

Reviewer #1: No

Reviewer #2: N/A

3. Have the authors made all data underlying the findings in their manuscript fully available (please refer to the Data Availability Statement at the start of the manuscript PDF file)?

Reviewer #1: Yes

Reviewer #2: Yes

4. Is the manuscript presented in an intelligible fashion and written in standard English?

Reviewer #1: Yes

Reviewer #2: Yes

5. Review Comments to the Author

Reviewer #1: “Economic and econometric methods to measure the illicit tobacco trade: a scoping review”

General comments and some key concerns:

It is an interesting study that is giving an insight on the review of the “Economic and econometric methods to measure the illicit tobacco trade”. Despite of tobacco being not allowed by law or custom or society in many countries, it is still traded and it provides a sizeable income to many governments in form of taxes. The present manuscript has reviewed the Economic and econometric methods to measure the illicit tobacco trade. However, there are some comments that need to be addressed as below.

• Authors should mind about the grammar and use of tenses when reporting in the manuscript

1. Abstract

• In the method section, the statement “Google Scholar for grey literature”, these two mean different things and therefore the authors need to separate them and not one leading to the other

• The authors used Google scholar only as a database, were there no other data bases used such as PubMed, Medline and many others. Didn’t this create publication selection bias and how was this overcome?

• Similarly, authors used publications in English, were there good relevant publications in other languages which were excluded? Again how was this overcome to avoid language bias?

• In the results section, the statement “23 used other economic or econometric models to assess tobacco or marijuana demand”. What were those other economic or econometric models?

• In the results section “marijuana” is mentioned and yet it was not in the title and aim of the review

• There are no results for the “trends” in the abstract as menti0oned in the conclusion

2. Introduction

• The introduction seems to be well written except a few issues including:

o On the illicit tobacco trade, what are the various tobacco products that are traded and does this include the products that are home grown?

3. Methods section

• Methods : Suggest it is changed to “Methodology”

• Line 2 in the methods, grammar issue that needs to be corrected. Authors should use past tense

• The inclusion criteria “Publications that examine the other illicit trade or related topics were included..”, I think these should not be included since they are out of scope. The implication of the findings can just be used in other relevant areas and to form a basis to do the same study focusing on those other areas

• The authors have not mentioned dissertations or theses that form the gray literature, couldn’t these also be useful. Isn’t there a publication bias?

• In the search strategy, PubMed and other databases were used but in the abstract only Google scholar is mentioned. Authors need to clear this!

• The key words that were used in the searching are missing and how they were combined. See the Boolean operators in the search strategy!!!

• The disagreement on which papers should be incorporated in the study, discussion among the 3 authors’ reviewers and agreement, was this the only based choice to go over it?

• How was the synthesized data summarized?

4. Results

• In the results section, were there regional distributions of the papers? And what was the contribution of those papers such as original papers, reviews, reports, dissertation/theses etc.

• When the” an ‘uplift factor” was used, there any differences in the outcomes when compared to the original model?

• CGA were excluded but the authors have not mentioned the other models that were used i.e. what are those Economic/econometric Models that were used for Critical Appraisal? What are their strength and drawbacks?

• Not all the research questions have been addressed i.e. where is the results for the trend and health equity?

5. Conclusion

• The authors should have conclusions all the research questions raised.

Reviewer #2: The manuscript presents a scoping review of economic and econometric methods to measure the illicit tobacco trade. The reviewed promises a lot and I was a little bit disappointed with the lack of depth in the findings.

The manuscript contains several typographical errors so would benefit from careful proofreading.

Background

Suggest breaking this down into more than one paragraph to enhance readability.

For the benefit of less specialist readers, I suggest saying a bit more about what is meant by economic/econometric methods / how these are defined, and justify the focus of the review on these over the other approaches. Suggest explaining more about what the CGA method is.

Methods

OSF – should be ‘Open Science Framework’.

* This point could do with more explanation – what sort of other illicit trade? Why was this deemed applicable to tobacco? Also, was the search designed to capture all relevant papers on other types of illicit trade?

The presentation of the eligibility criteria is a bit hard to follow with the multiple asterisks and final point in brackets. Could be revised for readability.

Data extraction – please provide details on what information was extracted from the papers. The lack of detail here is ultimately reflected in the findings – see my comment on this below.

Results

I found the results quite disappointing, in that they did not really address the main research question ‘What economic or econometric methods should be used to measure the size of the illicit tobacco trade?’ The findings were very descriptive, focussing more on what methods have been used, rather than the advantages and disadvantages of the different approaches. Given that this is a scoping review, I think the findings could have been more discursive, drawing on what the authors of those papers identified as strengths and weaknesses as well as identifying the methods themselves.

One of your research questions is around suitability of methods for assessing equity, but you say CGA cannot be used to investigate this - it is a bit unclear why this approach was nonetheless included in the review.

A summary table would be helpful, setting out the types of methods, key points about how they work, main strengths and weaknesses etc.

Discussion

The paragraph where you talk about the Likert scale was not clear.

You conclude by saying that triangulation is important – it would be helpful to provide some more concrete recommendations, e.g. by bringing some of the other approaches back into the discussion – surveys etc – and talking about what this triangulation should/could involve.

6. PLOS authors have the option to publish the peer review history of their article (what does this mean? ). If published, this will include your full peer review and any attached files.

**Do you want your identity to be public for this peer review?** For information about this choice, including consent withdrawal, please see our Privacy Policy .

Reviewer #1: No

Reviewer #2: No

---

## [Author Response · Author response to Decision Letter 1]

26 Jun 2025

Addressing the comments of the editor and reviewers

Thank you for the helpful comments and feedback; we have addressed them as follows:

Editor’s and reviewers’ comment Address Page

Editor’s comments

I noticed that the opposite of inclusion are written as exclusion criteria. Could you remove exclusion criteria (these need not necessary to be there). Deleted the exclusion criteria #6

If you had to exclude any articles among those included after title, abstract or full text review, please state the reasons. The reasons for exclusion were added. #6 and #7

Reviewer 1 comment

Authors should mind about the grammar and use of tenses when reporting in the manuscript

Thank you for the feedback, we have carefully reviewed (and edited) the text to address this point.

Abstract

In the method section, the statement “Google Scholar for grey literature”, these two mean different things and therefore the authors need to separate them and not one leading to the other

The authors used Google scholar only as a database, were there no other data bases used such as PubMed, Medline and many others. Didn’t this create publication selection bias and how was this overcome?

The ‘“Google Scholar for grey literature” text has been removed.

We used Google following the University of Otago’s method;

We searched six databases (

PubMed, CINAHL, EMBASE, EconLit, and ABI/Inform and MEDLine) and two economic working paper platforms (SSRN and IDEAS). These are key databases for public health and economics. Due to the limitations noted in the abstract, we could not add the details.

We have modified the text: “We searched six key databases for public health and economic papers (PubMed, CINAHL, EMBASE, EconLit, and ABI/Inform and Medline) and two economic working paper platforms (SSRN and IDEAS) for papers. Additionally, we searched for grey literature on Google and added the expert-selected articles.“

#2

Authors used publications in English, were there good relevant publications in other languages which were excluded? Again how was this overcome to avoid language bias?

We extensively searched the key public health and economics databases and added expert-suggested articles and grey literature. By doing this, we are confident there is minimal selection bias. We have discussed the details of this in the discussion section of the main article. #70 and #71

In the results section, the statement “23 used other economic or econometric models to assess tobacco or marijuana demand”. What were those other economic or econometric models?

We have added the models in the results section of the abstract as follows:

“The review included 39 studies: 16 applied Consumption GAP Analysis (CGA), and 23 used other economic or econometric models such as Hursh And Silberberg's Exponentiated Model, Discrete Choice Models, Extended Cost–Benefit Analysis and A Static Partial Equilibrium Model, Consumption Models, Consumption Models, Risk Prediction Models, A Forward-looking Behavioural Model, Integrated Micro-Macro Model of Demand, Endogenous Switching Regression Model, Multiple and Non-Linear Regression Model, Prevalence Projection Model, Demand-driven Analytical Model, Econometric Regressions Model, Econometric Modelling and two way fixed effects model. Only six of the 39 studies addressed health equity. “

#2

In the results section “marijuana” is mentioned and yet it was not in the title and aim of the review We only aimed to look at the illicit tobacco trade.

“Marijuana” is mentioned because of the selection criteria, which state “We included English publications from 2010 onwards that examined economic or econometric models assessing changes in illicit tobacco trade or related topics.”

We have clarified this in the discussion section of the main article.

#2, #70, #71

There are no results for the “trends” in the abstract, as mentioned in the conclusion This was inadvertently omitted but has been added now.

“CGA was used to estimate the existing illicit tobacco market size and assess the market trends over the past years, while the other models were used to assess and quantify the existing behavior and potential behaviors of engaging with different tobacco and other products, including illicit tobacco.”

In the conclusion section, we added

“Measuring the illicit tobacco trade is challenging due to its covert nature, methodological constraints, and limited high-quality data. Economic method or econometric method selection depends on the research objective: CGA is suitable for assessing national market trends but is limited in evaluating subpopulations or future policy impacts. Economic models are suitable for understanding or predicting user behaviour, including from a health equity perspective.”

#2 and #3

Introduction

The introduction seems to be well written except a few issues including: On the illicit tobacco trade, what are the various tobacco products that are traded and does this include the products that are home grown? To address the point raised we have added: “According to data collected from NZ Customs, many different types of illicit tobacco were seized during this period – illicit cigarettes, illicit loose tobacco imported from other countries (especially Tonga), chewing tobacco, water pipe tobacco, and domestically grown and produced loose tobacco (Bullen et al., 2023). Under the NZ Customs and Excise Act 2018, adults are permitted to cultivate up to 5 kg of tobacco annually for personal use but the sale or distribution of home-grown tobacco is prohibited and illegal (New Zealand Customs Service, 2023)“ #4

Methods:

Suggest it is changed to “Methodology” We have changed the heading to “Methodology” #6

Line 2 in the methods, grammar issue that needs to be corrected. Authors should use past tense Thank you for pointing this out. We have edited the text accordingly. #6

The inclusion criteria “Publications that examine the other illicit trade or related topics were included..” I think these should not be included since they are out of scope. The implication of the findings can just be used in other relevant areas and to form a basis to do the same study focusing on those other areas.

Thank you for the comment. We considered this point when we developed the protocol. However, we kept the criteria more open because we wanted to explore the economic and econometric methods as much as possible. Although our search strategy focused on tobacco, we included studies that examined illicit tobacco or other tobacco products or other illicit products like marijuana.

We have addressed this point in the discussion. #70 #71

The authors have not mentioned dissertations or theses that form the gray literature, couldn’t these also be useful. Isn’t there a publication bias?

We added all the publications, including theses or dissertations, if they met the inclusion criteria. We have modified the text of the eligibility criteria accordingly. #6 and #7

In the search strategy, PubMed and other databases were used but in the abstract only Google scholar is mentioned. Authors need to clear this! We mentioned that we searched six databases, plus Google for grey literature. Full details were not added given word count limitations.

However, we have now added to the abstract all the databases we searched. There was a typo error as we searched for grey literature on Google not on Google Scholar. #8

The keywords that were used in the search are missing, and how they were combined. See the Boolean operators in the search strategy The overall search strategy, as well as details for each database, were listed in the protocol; we have added the overall search strategy in the main manuscript as Table 1. #7 and #8

The disagreement on which papers should be incorporated in the study, discussion among the 3 authors’ reviewers and agreement, was this the only based choice to go over it?

Our discussion is based on the criteria, and the key criteria is the economic or econometric methods.

We have modified the text accordingly to make this clearer. #8

How was the synthesized data summarized?

The format to synthesize the data was detailed in the protocol. We have now added this information to the main manuscript as Table 2. #9

Results

In the results section, were there regional distributions of the papers? Yes - we have added this information to the paper. #10

And what was the contribution of those papers such as original papers, reviews, reports, dissertation/theses etc. We have added this information to the manuscript. #10

When the” an ‘uplift factor” was used, there any differences in the outcomes when compared to the original model?

We have added this information to the manuscript. #11

CGA were excluded but the authors have not mentioned the other models that were used i.e. what are those Economic/econometric Models that were used for Critical Appraisal? What are their strength and drawbacks? We have added the models used into the critical appraisal; our approach was to appraise from the health equity perspective of each study in Table 6. #62

Not all the research questions have been addressed i.e. where is the results for the trend and health equity?

Thank you, assessing the trend is not part of our research session but our findings from CGA as we went through the papers and methods. From the methods, we found out that CGA is useful for assessing the trends from estimating the illicit tobacco in the past, while other economic/econometric models could be applied to estimate the past or current illicit tobacco use behaviour or predict the future. We have assessed all the models from health equity perspectives. We have revised this in the results and in discussion sessions. #10,11,12 #30

#62 to #71

Conclusion

The authors should have conclusions about all the research questions raised.

We have added this information to the manuscript. #71 and #72

Reviewer 2

The manuscript presents a scoping review of economic and econometric methods to measure the illicit tobacco trade.

The reviewed promises a lot and I was a little bit disappointed with the lack of depth in the findings.

The manuscript contains several typographical errors so would benefit from careful proofreading.

Thank you for your review and comments.

We have summarised the models and main findings in the main paper. Two tables were placed into Appendix 2 and now we incorporated them into the main paper.

We have carefully proof-read the paper.

Background

Suggest breaking this down into more than one paragraph to enhance readability. We have broken the section into Additional paragraphs, as suggested. #4 and #5

For the benefit of less specialist readers, I suggest saying a bit more about what is meant by economic/econometric methods / how these are defined, and justify the focus of the review on these over the other approaches. Suggest explaining more about what the CGA method is.

These definitions were explained in the study protocol and attached in an appendix. We have now added them to the introduction. #4 #5

Methods

OSF – should be ‘Open Science Framework’.

Thank you. We have made this change. #6

* This point could do with more explanation – what sort of other illicit trade? Why was this deemed applicable to tobacco? Also, was the search designed to capture all relevant papers on other types of illicit trade? We have added the detail requested #6,#7 and #8

The presentation of the eligibility criteria is a bit hard to follow with the multiple asterisks and final point in brackets. Could be revised for readability.

We have revised the text accordingly. #6 and #7

Data extraction: please provide details on what information was extracted from the papers. The lack of detail here is ultimately reflected in the findings – see my comment on this below.

This information was in protocol, and appendices. We have now added the detail into the main manuscript. #9

Results

I found the results quite disappointing, in that they did not really address the main research question ‘What economic or econometric methods should be used to measure the size of the illicit tobacco trade?’ The findings were very descriptive, focussing more on what methods have been used, rather than the advantages and disadvantages of the different approaches. Given that this is a scoping review, I think the findings could have been more discursive, drawing on what the authors of those papers identified as strengths and weaknesses as well as identifying the methods themselves.

We have made major revisions to the results section to address this comment. We have added a summary table of the models and objectives with strengths and limitations.

#10 to #67

One of your research questions is around suitability of methods for assessing equity, but you say CGA cannot be used to investigate this - it is a bit unclear why this approach was nonetheless included in the review.

We have added the CGA to the equity assessment. #62

A summary table would be helpful, setting out the types of methods, key points about how they work, main strengths and weaknesses etc.

We have now done this #30 to #35

Discussion

The paragraph where you talk about the Likert scale was not clear We have revised the paragraph accordingly #67 to #71

You conclude by saying that triangulation is important – it would be helpful to provide some more concrete recommendations, e.g. by bringing some of the other approaches back into the discussion – surveys etc – and talking about what this triangulation should/could involve.

We have revised the conclusion section accordingly. #71, #71 and #72

---

## [Decision Letter · Decision Letter 1]

1 Oct 2025

PGPH-D-24-02979R1

Economic and econometric methods to measure the illicit tobacco trade: a scoping review.

Dear Dr. Phyo,

Thank you for submitting your manuscript to PLOS Global Public Health. After careful consideration, we feel that it has merit but does not fully meet PLOS Global Public Health’s publication criteria as it currently stands. Therefore, we invite you to submit a revised version of the manuscript that addresses the points raised during the review process.

We look forward to receiving your revised manuscript.

Kind regards,

Pranil Man Singh Pradhan, M.D.

Academic Editor

Journal Requirements:

1. Your manuscript is a Scoping Review. Please change the article type to 'Research Article', and ensure all submission questions are completed, noting the change in publication fees associated with Research Articles.

2. Please send a completed 'Competing Interests' statement, including any COIs declared by your co-authors. If you have no competing interests to declare, please state “The authors have declared that no competing interests exist”. Otherwise please declare all competing interests beginning with the statement “I have read the journal's policy and the authors of this manuscript have the following competing interests:”

For more information, please go to our submission guidelines:

https://journals.plos.org/globalpublichealth/s/submission-guidelines#loc-competing-interests

3. Please provide a complete Data Availability Statement in the submission form, ensuring you include all necessary access information or a reason for why you are unable to make your data freely accessible. If your research concerns only data provided within your submission, please write “All data are in the manuscript and/or supporting information files.” as your Data Availability Statement.

4. Please provide separate figure files in .tif or .eps format only and ensure that all files are under our size limit of 10MB.

5. Please include a separate legend or caption for Figure 1 in your manuscript.

6. We have noticed that you have uploaded Supporting Information files, but you have not included a list of legends. Please add a full list of legends for your Supporting Information files before or after the references list.

Additional Editor Comments (if provided):

Reviewers' comments:

Reviewer's Responses to Questions

**Comments to the Author**

1. If the authors have adequately addressed your comments raised in a previous round of review and you feel that this manuscript is now acceptable for publication, you may indicate that here to bypass the “Comments to the Author” section, enter your conflict of interest statement in the “Confidential to Editor” section, and submit your "Accept" recommendation.

Reviewer #3: (No Response)

Reviewer #4: (No Response)

2. Does this manuscript meet PLOS Global Public Health’s publication criteria ? Is the manuscript technically sound, and do the data support the conclusions? The manuscript must describe methodologically and ethically rigorous research with conclusions that are appropriately drawn based on the data presented.

Reviewer #3: Yes

Reviewer #4: Partly

3. Has the statistical analysis been performed appropriately and rigorously?

Reviewer #3: No

Reviewer #4: N/A

4. Have the authors made all data underlying the findings in their manuscript fully available (please refer to the Data Availability Statement at the start of the manuscript PDF file)?

Reviewer #3: Yes

Reviewer #4: Yes

5. Is the manuscript presented in an intelligible fashion and written in standard English?

Reviewer #3: No

Reviewer #4: Yes

6. Review Comments to the Author

Reviewer #3: Study should be align with the objectives and deviation of objectives to satisfy the data requirement can not be justified. As the objective and search strategy is for tobacco, discussion on other substances is causing as serious bias.

Reviewer #4: The scoping review is methodologically sound and rich in content, but it would benefit from transparency in data methods and appraisal. The limitations need to be highlighted clearly and make future recommendations based on the same.

Specific points to be considered by the authors:

• Kindly report both abstract and manuscript using the PRISMA-ScR checklist. In the methods section of abstract, include quality assessment.

• The review explicitly states that "no assessment was undertaken of the individual models employed" and that CGA studies were excluded from critical appraisal because the method is based on mathematical calculations and irrelevant to equity appraisal criteria. Thus, it can be mentioned clearly in limitation and applied in future recommendation.

• The review highlights that "only six of the 39 studies addressed health equity" and explicitly notes that CGA studies "did not reflect equity aspects" because it typically assesses illicit trade at the national level and lacks disaggregated data for subpopulations. Thus, review should also offer more concrete suggestions on how economic models could better integrate health equity components where currently lacking, beyond just stating their suitability.

• The inclusion criteria allow for studies on "other forms of illicit trade (e.g., illicit alcohol, illegal drugs, or various types of nicotine and tobacco products)". However, the population in the PCC framework is restricted to active tobacco users. This is also reflected in concept 5 of exclusion. While justified by the research team for exploring methods, the discussion does not delve into whether these methods are equally applicable or require significant adaptation for tobacco, or if these studies truly enriched the understanding of illicit tobacco trade beyond just methodological exploration.

• The exclusion of non-English language papers due to resource limitations is noted. The review could propose strategies for future reviews to mitigate language bias, such as collaboration with multilingual researchers, or a more targeted search for seminal works in specific non-English languages if resources allow.

7. PLOS authors have the option to publish the peer review history of their article (what does this mean? ). If published, this will include your full peer review and any attached files.

**Do you want your identity to be public for this peer review?** For information about this choice, including consent withdrawal, please see our Privacy Policy .

Reviewer #3: No

Reviewer #4: **Yes:** Reshu Agrawal Sagtani

Figure Resubmissions:

---

## [Decision Letter · Decision Letter 2]

29 Jan 2026

PGPH-D-24-02979R2

Economic and econometric methods to measure the illicit tobacco trade: a scoping review.

Dear Dr. Phyo,

Thank you for submitting your manuscript to PLOS Global Public Health. After careful consideration, we feel that it has merit but does not fully meet PLOS Global Public Health’s publication criteria as it currently stands. Therefore, we invite you to submit a revised version of the manuscript that addresses the points raised during the review process.

The manuscript has been evaluated by one reviewer, and the comments are available below.

The reviewer raised some concerns that need attention. For example, they would like you to be more general for nations where tobacco use and related harms disproportionately affect certain population sub-groups including socioeconomically disadvantaged and underserved communities.

Could you please revise the manuscript to carefully address the concerns raised?

We look forward to receiving your revised manuscript.

Kind regards,

Katrien G. Janin, PhD

Staff Editor

Journal Requirements:

Additional Editor Comments (if provided):

Reviewers' comments:

Reviewer's Responses to Questions

**Comments to the Author**

1. If the authors have adequately addressed your comments raised in a previous round of review and you feel that this manuscript is now acceptable for publication, you may indicate that here to bypass the “Comments to the Author” section, enter your conflict of interest statement in the “Confidential to Editor” section, and submit your "Accept" recommendation.

Reviewer #5: All comments have been addressed

2. Does this manuscript meet PLOS Global Public Health’s publication criteria ? Is the manuscript technically sound, and do the data support the conclusions? The manuscript must describe methodologically and ethically rigorous research with conclusions that are appropriately drawn based on the data presented.

Reviewer #5: Yes

3. Has the statistical analysis been performed appropriately and rigorously?

Reviewer #5: Yes

4. Have the authors made all data underlying the findings in their manuscript fully available (please refer to the Data Availability Statement at the start of the manuscript PDF file)?

Reviewer #5: Yes

5. Is the manuscript presented in an intelligible fashion and written in standard English?

Reviewer #5: Yes

6. Review Comments to the Author

Reviewer #5: The paper is in great shape. I have a few minor suggestions to help improve the paper.

Introduction

Page 5: In the statement “Economic methods have been used in tobacco control research for a variety of purposes: to study the relationships among taxation, price, consumption, and health outcomes; to model nicotine addiction within the framework of rational economic behaviour; to assess how cigarette taxation might shift demand toward other tobacco products; to evaluate the impact of advertising and advertising bans on cigarette demand; and to explore the effects of information dissemination, smoke-free laws, and the overall economic contributions of tobacco (14)”, I suggest changing the phrase “the overall economic contributions of tobacco” to “the economic costs of tobacco”.

Page 5: The statement specific to NZ “This focus was particularly relevant in the NZ context, where the prevalence of tobacco use varies significantly among different ethnic groups and socio-economic strata” can be made more general for nations where tobacco use and related harms disproportionately affect certain population sub-groups including socioeconomically disadvantaged and underserved communities.

Methodology

Page 8: In Table 2, CGA-related studies are separated from studies using economic or econometric methods. Do the authors consider that CGA-related studies are not economic studies? Consumption gap analysis involves valuation of the size of legal and illegal markets based on consumer demand for the products, which are very well subject of economic research. I would feel more comfortable to name all studies as using economic and econometric methods and divide them into two groups: CGA and non-CGA based studies.

Page 8: “We examined the strengths and limitations of these methods through a health equity lens, emphasising their relevance to priority populations rather than the quality of individual models.”—Who are included in the priority populations groups? A clear definition in this section would help readers better understand the relevance of the review to health equity perspective.

The PRISMA diagram in Figure 1 is blurry and can hardly be read.

Results

Page 10: “In one study (40)researchers utilised data from the ‘Euromonitor International’, as country-level data were unavailable(40) .” Is there any information in the study reviewed on how Euromonitor International estimate illicit market size?

Table 3: The column titled “Currency of calculations” makes one to expect that the market value of illicit tobacco is presented in the study which is true for some studies but not all. It is better reflected if you call it “unit of measurement”.

Since industry-funded studies were not excluded from the review, I would recommend reporting in a separate column the funder of the study to enable readers to identify industry-funded studies. It is also important to explain that industry-funded studies are prone to overestimate the size of illicit trade.

Discussion

Page 44: The limitation that the CGA estimates are calculated only at the national level is not exactly true. It is possible to apply the CGA method to subnational level to obtain estimates for specific geographic areas. See for example method of estimating illicit market size of commercial cannabis in multiple states in USA in:

Nigar Nargis, Samuel Asare, J Lee Westmaas, Measuring commercial cannabis availability: findings from a multi-state surveillance study in the US, Health Affairs Scholar, Volume 3, Issue 12, December 2025, qxaf209, https://doi.org/10.1093/haschl/qxaf209

The review found studies on illicit tobacco trade mostly do not cover the healthy equity perspective. From Table 6, it looks like a few studies (e.g., Tucker, 2017; Tangtammaruk, 2017, etc.) did have equity perspective. There is need for expanding the discussion on how these studies implemented the equity perspective and what lessons can be drawn from these studies for other approaches to studying illicit tobacco trade.

7. PLOS authors have the option to publish the peer review history of their article (what does this mean? ). If published, this will include your full peer review and any attached files.

**Do you want your identity to be public for this peer review?** For information about this choice, including consent withdrawal, please see our Privacy Policy .

Reviewer #5: No

 Figure Resubmissions:

---

## [Decision Letter · Decision Letter 3]

23 Feb 2026

PGPH-D-24-02979R3

Economic and econometric methods to measure the illicit tobacco trade: a scoping review.

Dear Dr. Phyo,

Thank you for submitting your manuscript to PLOS Global Public Health. After careful consideration, we feel that it has merit but does not fully meet PLOS Global Public Health’s publication criteria as it currently stands. Therefore, we invite you to submit a revised version of the manuscript that addresses the points raised during the review process.

We look forward to receiving your revised manuscript.

Kind regards,

Shashika Bandara

Academic Editor

Journal Requirements:

Additional Editor Comments (if provided):

Dear Authorship team,

Thank you for all the effort in revising this paper multiple times. I know it is time consuming and takes effort. The paper reads well and all reviewer comments have been adequately addressed. I have one minor suggestion to bring this paper methodologically on par with other reviews: this is to include a inclusion/exclusion criteria table. This will be a very quick revision given that you have inclusion criteria is outlined already. If you would like an example, you can look at this paper: https://academic.oup.com/heapro/article/39/6/daae155/7906019

Once you address that, I can proceed to accepting your paper. Thanks for the work which will help advance tobacco control efforts and inform policy.

Shashika

Reviewers' comments:

Reviewer's Responses to Questions

**Comments to the Author**

1. If the authors have adequately addressed your comments raised in a previous round of review and you feel that this manuscript is now acceptable for publication, you may indicate that here to bypass the “Comments to the Author” section, enter your conflict of interest statement in the “Confidential to Editor” section, and submit your "Accept" recommendation.

Reviewer #5: All comments have been addressed

2. Does this manuscript meet PLOS Global Public Health’s publication criteria ? Is the manuscript technically sound, and do the data support the conclusions? The manuscript must describe methodologically and ethically rigorous research with conclusions that are appropriately drawn based on the data presented.

Reviewer #5: Yes

3. Has the statistical analysis been performed appropriately and rigorously?

Reviewer #5: N/A

4. Have the authors made all data underlying the findings in their manuscript fully available (please refer to the Data Availability Statement at the start of the manuscript PDF file)?

Reviewer #5: Yes

5. Is the manuscript presented in an intelligible fashion and written in standard English?

Reviewer #5: Yes

6. Review Comments to the Author

Reviewer #5: Thank you for addressing my comments adequately.

7. PLOS authors have the option to publish the peer review history of their article (what does this mean? ). If published, this will include your full peer review and any attached files.

**Do you want your identity to be public for this peer review?** For information about this choice, including consent withdrawal, please see our Privacy Policy .

Reviewer #5: **Yes:** Nigar Nargis

 Figure Resubmissions:

---

## [Editor Report · Decision Letter 4]

27 Feb 2026

Economic and econometric methods to measure the illicit tobacco trade: a scoping review.

PGPH-D-24-02979R4

Dear Dr Phyo,

We are pleased to inform you that your manuscript 'Economic and econometric methods to measure the illicit tobacco trade: a scoping review.' has been provisionally accepted for publication in PLOS Global Public Health.

Best regards,

Shashika Bandara

Academic Editor

Thank you for adding the inclusion/exclusion criteria table. Congratulations on an impactful paper.